# Of pathogens and party lines: Social conservatism positively associates with COVID-19 precautions among U.S. Democrats but not Republicans

Theodore Samore[1,2]*, Daniel M. T. Fessler[1,2,3], Adam Maxwell Sparks[4], Colin Holbrook[5]

1 Department of Anthropology, University of California, Los Angeles, California, United States of America,
2 UCLA Center for Behavior, Evolution, and Culture, Los Angeles, California, United States of America,
3 UCLA Bedari Kindness Institute, Los Angeles, California, United States of America, 4 Department of Psychology, University of Guelph, Guelph, Canada, 5 Department of Cognitive and Information Sciences, University of California, Merced, California, United States of America

* theo.samore@gmail.com

**Data Availability Statement:** Complete surveys, datasets, analysis code, and preregistrations of predictions and methods are available on the Open Science Framework (https://osf.io/k92wg/).

## Abstract

Social liberals tend to be less pathogen-avoidant than social conservatives, a pattern consistent with a model wherein ideological differences stem from differences in threat reactivity. Here we investigate if and how individual responses to a shared threat reflect those patterns of ideological difference. In seeming contradiction to the general association between social conservatism and pathogen avoidance, the more socially conservative political party in the United States has more consistently downplayed the dangers of COVID-19 during the ongoing pandemic. This puzzle offers an opportunity to examine the contributions of multiple factors to disease avoidance. We investigated the relationship between social conservatism and COVID-19 precautionary behavior in light of the partisan landscape of the United States. We explored whether consumption of, and attitudes toward, different sources of information, as well as differential evaluation of various threats caused by the pandemic —such as direct health costs versus indirect harms to the economy and individual liberties —shape partisan differences in responses to the pandemic in ways that overwhelm the contributions of social conservatism. In two pre-registered studies, socially conservative attitudes correlate with self-reported COVID-19 prophylactic behaviors, but only among Democrats. Reflecting larger societal divisions, among Republicans and Independents, the absence of a positive relationship between social conservatism and COVID-19 precautions appears driven by lower trust in scientists, lower trust in liberal and moderate sources, lesser consumption of liberal news media, and greater economic conservatism.

## Introduction

In the spring and summer of 2020, the COVID-19 pandemic was profoundly shaping the personal, social, and political lives of most Americans. Although case counts across most of the

**Funding:** T.S., D.F., and A.S. benefited from U.S. Air Force Office of Scientific Research Award #FA9550-15-1-0137. T.S. benefited from support by the Templeton Religion Trust/Issachar Fund project "Science and Religion: An Evolutionary Perspective

**Competing interests:** The authors have declared that no competing interests exist.

United States were relatively low compared to the subsequent fall and winter waves, the pandemic's effects were already widely felt. Many people adopted a suite of prophylactic behaviors, including mask wearing, social distancing, disinfecting, and social isolation to avoid infection. Many businesses, services, and schools were ordered closed in order to stem the spread of the pandemic. In turn, the effects of the pandemic and subsequent closures resulted in substantial economic decline, sparking political debate about both the cost-benefit trade-offs of COVID-19-related restrictions, as well as the nature and extent of economic relief measures. Notably, the pandemic was also heavily politicized [1]. In general, politicians from the Republican party and sympathetic media figures downplayed the direct health severity of the pandemic relative to their Democratic counterparts, while emphasizing the costs of closures and restrictions to both the economy and personal liberty.

Potentially motivated by the lead-up to a presidential election in November 2020 and a desire to minimize a national crisis that could negatively impact his electability, then-President Donald Trump and his allies in the Republican party consistently downplayed the threat posed by the pandemic, claiming variably that the virus would disappear, that it was not any more dangerous than seasonal flu, and that prophylactic measures such as mask wearing were unnecessary [2]. Polling and research suggests that these attitudes among party elites were also reflected among supporters of the Republican party [3], including in their own health-related behaviors such as social distancing and mask-wearing [4, 5].

Yet, in contrast to this dynamic in the United States where Republicans–the more socially conservative party–have been more skeptical than Democrats of the dangers of the COVID-19 pandemic, a large previous literature has both theorized and demonstrated a positive association between social conservatism and sensitivity to threats, particularly threats from pathogens [see 6]. Here, in two studies conducted in the spring and summer of 2020, we explore the partisan patterns of U.S. responses to COVID-19 as a case study that challenges theoretical frameworks that link together political orientation, attitudes toward traditional norms, threat sensitivity, partisanship, and cost-benefit trade-off calculations between competing sources of threat.

In political psychology scholarship, *social conservatism* and *social liberalism* are largely treated as ends of an attitude spectrum representing, respectively, resistance to, or encouragement of social change [see 7]. An emerging body of theory and research suggests that individual differences in social conservatism are associated with individual differences in *threat sensitivity*–the tendency to process threat-related cues as salient, attention-garnering, emotionally evocative, and behaviorally motivating. At an ultimate level, such an association could occur if, over historical and evolutionary timescales, traditional social norms reliably mitigated threats. As a consequence, at the proximate level, and potentially independently of conscious awareness, individuals who are more sensitive to the possibility of threats may assume that traditional and socially normative practices offer a form of precaution and threat management.

The potential threat-mitigating properties of *traditions*—that is, practices and norms that invoke both a moral valence and a real or imagined time-depth—could manifest via multiple, non-mutually exclusive benefit streams. First, specific traditions may actually provide direct protection against threats; cultural evolution may produce norms that instrumentally mitigate the costs of either specific threats, or certain domains of threat, including pathogens [8]. Individuals might explicitly recognize or implicitly assume the specific connections. However, because the functionality of norms is frequently opaque to adherents [9, 10], hazards may be implicitly assumed to be addressed by endorsing traditions broadly, even in domains apparently unrelated to a given class of threats. This holds true so long as the frequency and magnitude of those instrumentally threat-mitigating traditions outweigh the potential costs of following non-instrumentally adaptive traditions as part of a wholesale commitment to

traditions broadly (although traditions can have adaptive value outside of their instrumentality, see below).

Second, via increased social support, adhering to traditional norms can provide broad benefits, including the mitigations of threats (such as coalitional conflicts, interpersonal conflicts, or illnesses), for example, by advertising the adherent's identity as a member of the in-group who merits aid [11, 12]. In addition, traditions may be felicitous as coordination devices.

Alternatively, a non-adaptive association between traditionalism and threat sensitivity could arise in particular cultural contexts, for example if individuals and/or institutions may present traditions as possessing threat mitigating properties or as being broadly beneficial, irrespective of the actual instrumental utility of those traditions. Via processes of cultural transmission, individuals may then ascribe threat mitigating benefits to those traditions.

If, over evolutionary time, on average these functions (or other unrecognized functions) resulted in traditions mitigating the costs of threats, then, as part of their evolved psychology, individuals may instinctively perceive that traditions writ large ameliorate the costs of those threats, either within or outside of conscious perception. However, adherence to traditions also entails costs, as, while subject to uncertainty, non-traditional practices can present valuable opportunities. The propensity to cleave to tradition or adopt innovations may thus partly hinge on whether, for a given individual in a given place and time, threats loom larger than opportunities.

In the U.S. and other large democracies, socially conservative political ideologies and parties frame their positions on social issues as maintaining the values and practices of the past [13]. Variation in threat sensitivity may thus shape political behavior and party preferences in countries such as the U.S. [6, 14], with higher threat sensitivity associated with both greater traditionalism and greater social conservatism. When considering associations between threat avoidance and socially conservative attitudes (encompassing both general attitudes toward traditions, and specific policy preferences that emphasize social continuity), it is important to distinguish distinct dimensions of political orientation. Social and economic conservatism reflect different ideological foci, where the former concerns attitudes toward social change, and the latter concerns attitudes toward fiscal policy [15]. Although these ideological dimensions sometimes cohere in contemporary political entities—such as the Republican party in the U.S. —the association may not be inherent. Assuming that social conservatism centers on maintaining the (real or imagined) practices of the past, while economic conservatism does not, the *traditional-norms account* laid out above privileges the former as the driver of the association between threat sensitivity and political ideology in general, implying that social conservatism should be more strongly related to threat avoidance than other forms of political ideology and related attitudes.

A growing empirical literature has tested this hypothesized link between threat sensitivity and attitudes toward social change, finding that preferences for tradition—as well as socially conservative political ideologies—associate with greater sensitivity toward certain threats [see 6, 16]. Convergently, evolutionary modeling work has demonstrated that, at the group level, high degrees of objective threat favor the evolution of greater norm adherence [17]. However, the extent to which sensitivities to different categories of threats are associated with political ideology is contested [18], and the volume of evidence varies by threat domain. Pathogen threat is one of the most extensively studied domains, and socially liberal, less traditional individuals have consistently been found to be less pathogen-avoidant than their conservative counterparts [19–21]. In sum, if greater threat sensitivity leads to upregulated threat-mitigation behaviors across different domains, then the general endorsement of traditions should tend to co-occur with other investments in threat mitigation such as pathogen avoidance behavior [19, 22].

Concordantly, a related body of scholarship has theorized and empirically tested the possibility of conservative shifts in response to real-life threats [14, 23–25; but see 26]. The underlying functional logic is shared with the traditional norms account: if traditions and socially conservative norms can mitigate the costs of recurrent threats, then cues of increased threat may lead individuals to flexibly upregulate their traditionalism and social conservatism in response. Indeed, the possibility that temporal variation in threats results in conservative shifts is not mutually exclusive with the possibility that trait threat sensitivity influences social conservatism. Rather, both relationships may result from a shared underlying process that links threat to resistance to social change.

In hypothesizing a link between socially conservative attitudes and threat sensitivity, much of the research examining the postulated relationship employs hypothetical scenarios that often unrealistically ignore trade-offs accompanying actual behavior. Such measures present threat cues, offering participants some basis for estimating the costs of exposure to threats, and therefore the benefits of threat-avoidance, but generally leave unspecified the costs of avoiding the threats (e.g., opportunity costs, increased vulnerability to other threats, etc.). Thus, prior research [6, 19, 21, 27] has focused mostly on the benefits of threat avoidance, in turn limiting the ecological validity of the findings. Greater attention to costs is needed to more fully understand cost-benefit tradeoffs. Accordingly, recent work has started to address the effects of tradeoffs on threat and pathogen avoidance behaviors [28, 29]. In addition, the previously discussed empirical observations of conservative shifts in response to real-world threats likely implicitly summarize the cost-benefit trade-off calculations that individuals may be making.

The present research seeks to address many of the limitations found in the prior literature. The COVID-19 pandemic involves a dangerous pathogen threat, one that is both highly salient for much of the world's population, and has had marked effects on real behavior. The extent to which variation in individuals' costly prophylactic responses associate with variation along the social and political dimensions discussed above may therefore illuminate the hypothesized relationship between socially conservative attitudes and threat sensitivity. Specifically, precautions taken in response to the COVID-19 pandemic may reflect threat sensitivity in the pathogen domain in light of the real-world trade-offs between different threat domains. Indeed, speaking to the possibility of these kinds of trade-offs, initial evidence suggests that individuals make COVID-19 precaution trade-offs with the mate-seeking domain [28]. Further, because reports of actual behavior summarize many of the implicit calculations being made by individuals, COVID-19 precautions plausibly more accurately reveal the intersection of baseline threat sensitivity and trade-offs among multiple threat domains than do questions about hypothetical cues of pathogen presence, generic items about danger, or broad statements regarding concern about the pandemic.

Some of the precautions recommended or required by public health authorities interfere with engaging in traditional practices (e.g., social distancing precludes family gatherings, public sporting events, in-person religious services, etc.). Accordingly, two possibilities exist regarding the relationship between threat sensitivity, socially conservative attitudes, and COVID-19 precautions. On the one hand, if more threat-sensitive individuals focus on the danger posed by COVID-19 over and above the conflict with various traditions entailed by precautionary behaviors, then they will report both greater precautionary behavior and greater valuation of traditions than will less threat-sensitive individuals. On the other hand, highly threat-sensitive individuals may view such behaviors as threats in themselves, endangering individual liberties or economic prosperity. If more threat-sensitive individuals focus on the precautions themselves, construing these as infringing on traditions, then they will report lower precautionary behavior than less threat-sensitive individuals, potentially resulting in a *negative* relationship between traditionalism and precautionary behaviors.

Initial empirical work suggests that social conservativism associates with precautions in response to COVID-19, and—relating back to the question of conservative shifts—that socially conservative attitudes may have increased since the start of the pandemic as a function of perceived threat [30–32; but see 33, 34]. Yet, in the U.S., the politicization of COVID-19 has resulted in supporters of the Republican party—despite being characterized by higher social conservatism and a stronger commitment to traditional cultural values [15, 35]—taking a more skeptical view of the danger posed by the disease [3], enacting fewer real-life precautions [5], and holding more negative attitudes toward precautions such as mask wearing [36]. This suggests either that the traditional-norms account is incorrect, or that other factors are influencing Republicans—either trade-offs with attempts to mitigate other threats, conflicts with traditions, or broader factors that are independent of threat mitigation per se. For example, different information environments—as a function of the types of media individuals consume, and the types of figures whom individuals trust—may relate to the specific threat cues that are experienced as most salient [37]. A primary objective of this study is thus to examine the relative contributions of political ideology and the consumption and endorsement of partisan messaging in relation to precautionary COVID-19 behaviors.

Here, we study whether more socially conservative and traditionalist individuals are more pathogen-avoidant in the face of a real-world disease threat unprecedented in recent memory, examining whether individual differences in ideology reflect precautionary behaviors, and assessing whether such relationships are associated with exposure to politicized messaging. Because contemporary American political parties are amalgams of different ideological dimensions, we attempt to disentangle perceived trade-offs between different threat domains, and the relationships among those various components—specifically, if economic conservatism is associated with greater salience of threats to economic liberties, and social conservatism has such a relationship with pathogen threats, then there may be a complex interplay between economic and social conservatism. Accordingly, exploratory analyses of our data can inform hypotheses about causal pathways among these variables.

Conducted in the context of the COVID-19 pandemic and the socio-political response in the U.S.A., our work tests the hypothesized connection between threat sensitivity and political beliefs, affording examination of these relationships and their connection to high stakes real-world behaviors. This novel context necessarily requires the use of untested measures, hence these studies are best considered preliminary. Below, we specify the issues examined.

## Research questions and hypotheses

### 1 How does COVID-19 precautionary behavior relate to socially conservative political dimensions?

The traditional-norms account of the relationship between socially conservative attitudes and threat sensitivity holds that, because traditions promise buffers against the vagaries of a dangerous world, individuals for whom particular threats–including pathogen threats–loom large will cleave more strongly to traditions. If so, and if COVID-19 prophylaxis indexes such dispositional greater threat sensitivity in the pathogen domain, then, all else held constant, socially conservative political attitudes should be associated with COVID-19 prophylaxis. The absence of such a relationship could be attributable to the hypothesis being wrong, or to a violation of the ceteris paribus assumption; we discuss the latter below. Alternatively, if people view some COVID-19 precautions as violating tradition, then precautions and socially conservative attitudes may negatively correlate, again violating the ceteris paribus assumption.

## 2 Do partisan differences play a role in the relationship between precautionary behaviors and socially conservative political differences?

The ceteris paribus assumption underlying the traditional-norms hypothesis may not apply. In the U.S., people receive different information about the pandemic as a function of media partisanship. For example, most of the public voices questioning the severity of the outbreak are conservative leaders and conservative media outlets associated with the Republican party [38, 39]; correspondingly, Republicans report less concern than Democrats that COVID-19 poses a major health threat [3]. Further, Republicans may weigh the economic and personal-liberty threats posed by prophylactic reactions to the pandemic as more serious relative to direct health threats. This suggests that, in the U.S., endorsement of socially conservative political attitudes and support for socially conservative political coalitions may not be associated with greater COVID-19 prophylaxis.

While there are principled reasons to think that perceived trade-offs between different domains may shape partisan differences in costly COVID-19 precautionary behaviors, many mechanisms could drive such differences. We approach possible countervailing drivers of partisan differences in COVID-19 precautions using a theoretically-motivated inductive approach, and include a large number of variables that may shape partisan differences in responses to COVID-19; below, we explain the rationale for each.

**2A) Accounting for other dimensions of ideological attitudes.** The traditional-norms account specifically predicts that socially conservative attitudes should associate with threat sensitivity, but does not make predictions regarding other dimensions of ideological attitudes, such as opinions concerning economic or militaristic political issues, or related personality traits such as social dominance orientation, authoritarian aggression, and submission to authority. Yet, these different facets of political belief are highly correlated [7], such that they need to be ruled out as causes of any relationship between socially conservative attitudes and precautionary COVID-19 behaviors.

Further, despite higher-order correlations, distinct ideological dimensions may lead individuals to differentially prioritize clashing threat domains. For example, economic conservatism—and preferences for limited government intervention in the public sphere—may heighten sensitivity toward perceived threats posed by government responses to the pandemic, such as public health directives and economic closures. These concerns may outweigh the perceived pathogen threat posed by COVID-19, or mask the relationship between socially conservative attitudes and pathogen avoidance behaviors [40].

Additionally, although a large literature indicates that conservatives are more threat-sensitive than liberals across many domains and mechanisms [6], some evidence suggests that conservatives may also view those threats—which they perceive more readily—as more easily vanquished [41]. In the context of the pandemic, this suggests that, while conservatives may recognize that the disease represents a substantial threat, they may also be more confident in their ability—or the ability of their leaders—to mitigate that threat. This may be related to confidence in one's traditions, or to aspects of right-wing authoritarianism and explicit political ideology, such as authoritarian aggression, submission to authority, and militaristic political orientation.

**2B) Media consumption habits.** Media outlets in the U.S. have covered COVID-19 from different perspectives, with conservative media being more skeptical of the severity and health consequences. The content of news coverage has been shown to shape both beliefs about scientific claims in general [42], and responses to COVID-19 in particular [43]. Thus, asymmetry in partisan coverage of the coronavirus outbreak may influence both precautionary behaviors and the relationship between those behaviors and socially conservative attitudes. Alternatively,

individuals' media choices may reflect, rather than cause, abiding differences that drive potential variation in responses to the pandemic along partisan lines; our data will not adjudicate between these possibilities.

In addition to differing in their exposure to various news streams, individuals differ in whom they listen to for advice about the outbreak, including media, political, and scientific sources. These differences are likely also important in determining how individuals' social preferences inform their behavioral responses to COVID-19, especially given partisan differences in the types of authorities that individuals trust.

**2C) Demographics.** Republicans and Democrats differ, on average, along multiple demographic dimensions, including ethnicity, gender, age, and education; they also live in regions differing in population density, creating differences in opportunities for disease transmission [44]. Additionally, at the time data were collected, the distribution of coronavirus outbreaks across the U.S. was highly skewed along geographical and urban/rural lines. We therefore explore whether such demographic variables influence the relationship between precautions and conservatism.

### 3 Are behavioral responses to COVID-19 related to trait pathogen avoidance?

An extensive corpus links pathogen avoidance to disgust sensitivity [45]. If disgust sensitivity is an emotion-potentiating mechanism for motivating some pathogen avoidance behaviors, then it should positively correlate with actual prophylaxis. Further, disgust sensitivity associates with socially conservative attitudes, per the traditional-norms account. Because disgust proximately motivates pathogen avoidance, disgust sensitivity measures may statistically account for the relationship between socially conservative attitudes and COVID-19 behaviors. Alternatively, pathogen disgust may not be as reliably triggered by cues of respiratory infection compared to other pathogen cues, in which case disgust may not mediate a precautions-socially conservative attitudes relationship.

## Methods

Studies were approved by the UCLA Office of the Human Research Protection Program. Written informed consent was obtained before participation. Complete surveys, datasets, analysis code, and preregistrations of predictions and methods are available at https://osf.io/k92wg/. The full measures can also be found in S1 Appendix.

### Project overview

A pilot study was conducted on April 17[th], 2020 to examine the hypotheses and develop measures (see S2 Appendix for full description of methods and results). Methods were subsequently refined. We enlarged sample size to enhance power for detecting effects of interest; increased the granularity of measures of media consumption, and of trust in individual and institutional information sources; and added detailed measured of responses to, and perceptions of, economic costs of the pandemic, as well as perceived threats to individual liberties posed by government mandates. We then ran two studies using identical methods 43 days apart; these conceptually replicated and extended the principal results of the pilot. Study 2 tested the replicability of Study 1, particularly given changes in the pandemic that could affect relationships between American political attitudes and precautionary behavior. For example, in the period between Studies 1 and 2, disease prevalence increased in U.S. regions that were less liberal than the urban areas which first saw large outbreaks [46], while individual and governmental precautions, such as mask wearing, became more politicized.

## Sample size

In the pilot study ($N$ = 433), socially conservative attitudes were correlated with COVID-19 precautionary behaviors at $r$ = .11 across all participants. This effect is consistent with previous studies examining the relationship between pathogen-threat sensitivity and similar political and attitudinal measures [19, 21]. With α = .05, and power = .80, the projected sample size needed given the pilot results is approximately 646. However, because we are interested in political party-specific effects, adequate subsamples for each major American political party are needed. In the pilot, 27% identified as Republican and 52% as Democratic. To sufficiently recruit in each subgroup to detect effect sizes consistent with the pilot, we doubled our recruitment target to approximately 1,000.

## Participants

In both Studies 1 and 2, 1,008 adult U.S. participants were recruited on Amazon Mechanical Turk and paid $2.75 (20-minute HIT, 99% approval rating, minimum completed HITs = 500). From the pilot study onward, each sample comprised workers who had not previously participated in this project. Data were prescreened for repeat participation, minimum completeness, minimum completion time, and answers to "catch questions". Study 1's final N was 906 (43% female; 69% white; age range 18–77 [$M$ = 39.2, SD = 12.2]). Study 1 ran on May 29th, 2020, when many lockdown orders were expiring across the U.S. [47]. Study 2's final N was 906 (49% female; 76% white; age range 18–89 [$M$ = 40.6, SD = 13.2]). Study 2 ran on July 11th, 2020, when cases were increasing in many U.S. states and were more widely geographically distributed, while lockdown orders varied widely [47] and some precautionary behaviors—such as mask wearing—had become more politicized [48].

## Measures

Measures, and the order of presentation, were identical in Studies 1 and 2. The order of the first four measures described below was randomly counterbalanced.

**Political orientation.**   Although political orientation is often described as if it were inherently unidimensional, such apparent unidimensionality may actually reflect partisan coalitional dynamics. Accordingly, rather than assume that individuals' positions are necessarily uniform across multiple components of political orientation, we measured political orientation using a modification of Dodd et al.'s [49] version of Wilson and Patterson's [50] multifaceted issues index. Participants were asked to indicate whether they agree, disagree, or are uncertain about various prominent issues in contemporary American politics. These are subdivided into three categories: social (e.g., abortion), economic (e.g., tax rates), and military (e.g., foreign intervention) issues. Agreement was scored as +1, disagreement as -1, and uncertainty as 0; liberal items were reverse scored, hence increasing positive values reflect greater conservatism. Responses were averaged within each subscale, producing a composite measure for each of the three dimensions.

**Traditionalism and right-wing authoritarianism.**   Participants completed the Aggression-Submission-Conventionalism scale (ASC), which measures the concepts of right-wing authoritarianism employing politically and religiously neutral language [51]. Here, we operationalized the Conventionalism subscale as reflecting attitudes toward traditions, as the items in this subscale are explicitly intended to measure, "commitment to the traditional social norms in one's society" [51], e.g., "Traditions are the foundation of a healthy society and should be respected". Participants also completed the *authoritarian aggression* (e.g., "Strong force is necessary against threatening groups") and *submission to authority* (e.g., "We should believe what our leaders tell us") subscales. For all three subscales, participants rated their

agreement with statements on a 7-point Likert scale, from "strongly disagree" to "strongly agree". Half of the items indicated agreement with traditionalism, aggression, and submission, and the other half (reverse scored) indicated disagreement. Scores were averaged within each subscale.

**Socially conservative attitudes.** Political orientation is often measured using a single-item unidimensional scale ranging from conservative to liberal. However, as we have noted, it is critical to separate distinct dimensions of ideology [14], such as economic and social conservatism or liberalism. Further, political ideology is complex, and encompasses both specific policy preferences in a given political context, as well as the kinds of general attitudes that help shape those preferences; in the context of social conservatism, the endorsement of tradition is likely a constituting attitude of the ideology. To operationalize social conservatism in light of these considerations—characterized by both specific policy preferences involving matters of tradition and cultural change, and general attitudinal orientation toward tradition and change—we created a composite socially conservative attitudes ideology scale. This composite scale consisted of the rescaled responses from the Dodd-style issues index and the conventionalism subscale of the ASC (see previous sections for example items). Both the issues index and the ASC scale have been widely used to measure social conservatism and attitudes toward tradition [e.g., 19, 49]. Further, because these individual scales focus on, respectively, general attitudes toward tradition, and specific policy preferences related to social conservatism, combining them provides a more complete measurement of socially conservative ideology. The resultant composite socially conservative attitudes variable was measured on a -1-to-1 scale, where increasing scores indicate increasing socially conservative attitudes. This composite was reliable ($\alpha$s = .89 –.90).

**Social dominance orientation.** We used the four-item Short Social Dominance Orientation Scale [52]. Participants rated agreement with items such as "Superior groups should dominate inferior groups" on a 7-point Likert scale, from "strongly disagree" to "strongly agree". Half of the items were reverse coded; responses were averaged across items.

**Pathogen disgust sensitivity.** Participants completed the pathogen subscale of the Three-Domain Disgust Scale [53], rating how disgusting they found seven hypothetical scenarios (e.g., stepping on dog feces) using a 7-point Likert scale, from "not disgusting at all", to "extremely disgusting"; responses were averaged across items.

**COVID-19 precautionary behaviors.** Our novel measure consisted of 12 questions concerning precautionary health behaviors in response to COVID-19, including the frequency of mask wearing, hand washing, social distancing, and disinfecting, and the importance to the participant of stocking up on supplies such as hand sanitizer and household disinfectants. Items were rated on 7-point scales, from either "never" to "as often as possible", or from "not important at all", to "extremely important". Participants were also asked the extent to which they were following local lockdown restrictions, and whether they had been careful to physically distance from people outside their household. Responses were averaged across items, creating a reliable composite ($\alpha$s = .85 –.86).

**Trust for sources of COVID-19 information.** Employing neutral language, we examined participants' confidence in various sources of information across the ideological spectrum about COVID-19. In Study 2, we included a range of individual media figures, identified by name only (e.g., conservative talk-radio host Rush Limbaugh), health professionals (e.g., Dr. Anthony Fauci, Director of the U.S. National Institute of Allergy and Infectious Diseases), politicians (e.g., U.S. President Donald Trump), media organizations (e.g., The New York Times), health organizations (e.g. the U.S. Centers for Disease Control and Prevention), and categories of people (e.g., liberal/conservative journalists, medical scientists). Using exploratory factor analysis (see S3 Appendix for details) to determine the structure of trust responses, we

extracted three conceptually coherent factors, which we labeled: *trust in scientists* (including items such as trust in Dr. Fauci and the CDC), *trust in liberal and moderate information sources* (including items such as trust in liberal and moderate journalists, and media figures such as liberal television news host Rachel Maddow), and *trust in conservative information sources* (including items such as trust in conservative journalists, and figures such as Rush Limbaugh). When averaged into separate composites, these factors were reliable in both studies (trust in scientists: αs = .86 –.87; trust in liberal and moderate information sources: αs = .95 –.96; trust in conservative information sources: αs = .96).

**Other COVID-19-related items.** We surveyed participants about various beliefs and experiences regarding the COVID-19 pandemic, divisible into six categories.

1. Perceived effectiveness of prophylactics against COVID-19: On a 7-point Likert scale from "not at all protective" to "extremely protective", participants rated the effectiveness in protecting against COVID-19 of a variety of prophylactics (e.g., "How well do you think each of the following protects you from COVID-19. . . hydroxychloroquine . . . mask-wearing?", etc.).

2. COVID-19 domain-specific threat-assessments: Participants gauged the relative hazards posed by different threats caused by the COVID-19 outbreak. We measured how participants weighed the perceived threat of the direct health hazards posed by the disease relative to two possible downstream costs of protective behavior: the economic fallout of the pandemic, and the perceived loss of personal liberties resulting from public health directives. Using 1 to 7 Likert-type scales, health-domain items included concern about contracting and spreading COVID-19 (e.g. "How concerned are you about. . .. Personally getting COVID-19 . . . Transmitting COVID-19 to a family member"), estimates of the health risks posed by infection (e.g., "How severe would the consequences of catching COVID-19 be to. . . your own health"), as well as questions regarding whether participants thought the threat of the pandemic was overblown, or would quickly pass (e.g., "Please indicate how strongly you agree or disagree with each of the following statements . . . I think that the threat of COVID-19 is overblown"). Economic and personal liberty-related items included self-reported concern over those issues (e.g., "How concerned are you about . . . losing personal liberties because of COVID-19 lockdown orders"), focus on defending personal liberties (e.g., "During the COVID-19 outbreak, how focused are you on doing the following . . . speaking out to defend personal liberties"), efforts to acquire guns and ammunition (e.g., "Within the last 10 weeks, it has been important to me that [I/my household] make an effort to stock up on . . . guns and ammunition"), and beliefs that the economic and personal liberty costs of the pandemic outweighed the health ones (e.g., "Please indicate how strongly you agree or disagree with each of the following statements . . ..I think that the economic costs of the COVID-19 response outweigh the public health benefits"). We created a reliable COVID-19 domain-specific threat-assessments composite based on these items (αs = .89 –.90). Health domain items were reverse scored, such that higher scores indicated finding the direct health consequences of the pandemic less serious, particularly in contrast to downstream threats to personal liberties and the economy.

3. Economic precautions: Participants were asked about the extent to which they were preparing for an economic downturn (e.g., "During the current COVID-19 outbreak, how focused are you on doing the following . . . reducing discretionary spending"). We averaged these behavior items into a composite scale, which was reliable (αs = .75 –.78).

4. Perceived prevalence of COVID-19: Participants gauged COVID-19 prevalence within their local communities, including their estimates of the current incidence, their neighborhood's density, and how many people they knew who had contracted COVID-19.

5. Political leadership assessments: Participants provided a series of assessments on a 1 to 7 Likert scale from "worst possible response" to "best possible response" about the effectiveness of the President, Congress, and the participant's state and local governments in their responses to the COVID-19 pandemic.

6. Additional items: As single items, participants also rated their perceived likelihood of contracting COVID-19 (e.g., "How likely do you think the following people are to become ill with COVID-19 . . . myself), the severity of the economic consequences they faced as a result of the pandemic (e.g., "How severe are the current economic consequences you face because of the COVID-19 outbreak?"), and their concerns about being able to access healthcare (e.g., "How concerned are you about . . . needing to seek in-person medical care for non-COVID related reasons?"). Additionally, if participants engaged in prophylaxis, they indicated whether those behaviors were primarily motivated out of concern for their own health or that of others, (e.g., "How much do you engage in these protective behaviors out of concern for your own health?"). Finally, participants were asked to indicate whether they had been infected with COVID-19, and, if so, whether they were still ill.

**News consumption.**   Participants indicated hours per week spent consuming news of any kind, then frequency (on a 1 to 7 Likert scale, from "never", to "very frequently") with which they attended to specific news outlets with unambiguous partisan leanings. Using Allsides Media Bias ratings, we assigned each news source to one of three composite measures based on its externally rated partisan lean: *liberal-leaning media consumption* (e.g., MSNBC; αs = .88 –.89), *moderate-leaning media consumption* (e.g., USA Today; αs = .65 –.68), and *conservative-leaning media consumption* (e.g., Breitbart; αs = .87 –.89). Because the moderate-leaning composite was unreliable, it was dropped from analysis.

**Endorsement of public health interventions.**   To measure opinions about a government public health intervention outside of the pandemic context,—we gauged participants' agreement with the government's intervention in tobacco use using four face-valid items, rated on a 1 to 7 scale, and averaged into a reliable composite (αs = .78 –.80).

**Demographics and study checks.**   Participants indicated their gender identity, ethnicity, age, belief in God or other deities, income, education, and preferred U.S. political party.

## Results

All analyses of scale variables make the simplifying assumption that Likert scale data can be treated as interval.

### Does COVID-19 precautionary behavior differ by political party?

After applying prescreening criteria, there were 906 participants in both Studies 1 and 2. In Studies 1 and 2, respectively, 424 and 413 participants identified as Democrats, 212 and 210 as Republicans, and 228 and 237 as Independents. Remaining participants—42 in Study 1, and 46 in Study 2—identified as members of the Green party, Libertarian party, or other. Because there were few self-identified supporters of the Green, Libertarian, and other American—precluding reliable detection of the effects of interest—they were excluded from analyses looking at party-specific effects. Given that these supporters are also at low frequency in the U.S., excluding these participants should not substantially impact the generalizability of results.

Examining Democrats, Republicans, and political Independents, there was a significant effect linking party affiliation to levels of precautionary behavior in both studies (Study 1: $F$[2, 860] = 12.8, $p$ = < .001; Study 2: $F$[2, 857] = 12.8, $p$ < .001). Post hoc comparisons using the Tukey HSD test indicate that the mean precaution scores for Democrats (Study 1: $M$ = 5.18, $SD$ = 1.02; Study 2: $M$ = 5.22, $SD$ = .98) were significantly higher than those for Republicans (Study 1: $M$ = 4.81, $SD$ = 1.24, $p$ < .001; Study 2: $M$ = 4.80, $SD$ = 1.28, $p$ < .001) and Independents (Study 1: $M$ = 4.77, $SD$ = 1.19, $p$ < .001; Study 2: $M$ = 4.91, $SD$ = 1.08, $p$ = .001), but that precautions did not significantly differ between Republicans and Independents (Study 1: $p$ = .921; Study 2: $p$ = .489).

## Do socially conservative political attitudes predict precautionary behavior?

Using linear regression with moderation, in both studies, COVID-19 prophylaxis associated with socially conservative political attitudes among Democrats, but not Republicans or Independents (Fig 1). Simple slopes analyses were performed to assess the conditional effects of socially conservative attitudes on precautions by political party. In both studies, these analyses showed that the conditional effects were significant among Democrats (Study 1: $B$ = .82, SE = .16, $t$(857) = 5.08, $p$ < .001; Study 2: $B$ = .74, SE = .15, $t$(854) = 4.98, $p$ < .001), but not Republicans (Study 1: $B$ = .02, SE = .24, $t$(857) = .09, $p$ = .939; Study 2: $B$ = -.05, SE = .21, $t$(854) = -.24, $p$ = .809) or Independents (Study 1: $B$ = .13, SE = .20, $t$(857) = .64, $p$ = .520; Study 2: $B$ = -.04, SE = .18, $t$(854) = -.23, $p$ = .818). That is, more socially conservative Democrats reported greater COVID-19 precautions relative to more socially liberal Democrats, however this relationship did not obtain among Republicans or Independents. Slopes did not significantly differ between Independents and Republicans (Study 1: $B$ = .11, SE = .31, $t$(857) = .35, $p$ = .728; Study 2: $B$ = .01, SE = .28, $t$(854) = .04, $p$ = .969). In sum, precautionary behavior was predicted by social conservatism among Democrats to a significantly greater extent relative to Republicans or Independents, who did not differ in this regard. This full pattern of results obtained in the Pilot Study as well (see S2 Appendix).

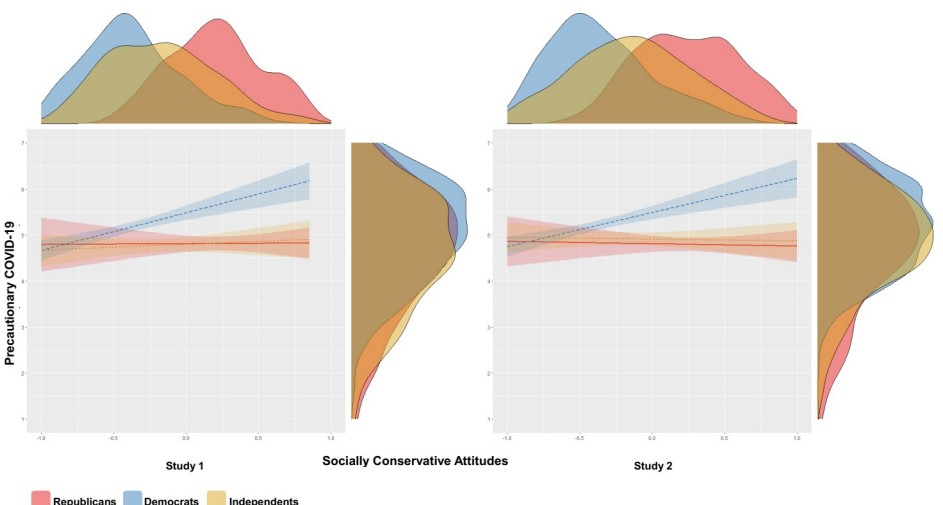

**Fig 1. Relationship between socially conservative attitudes on COVID-19 precautions is moderated by political party.** Studies 1 and 2 conditional effects of moderated linear regressions in which COVID-19 precautions were regressed on the (centered) socially conservative attitudes composite, political party affiliation, and their two-way interaction. Bands around regression lines are 95% confidence intervals. The density plots along the x-axes represent the raw distributions of socially conservative attitudes by political affiliation. The density plots along the y-axes represent the raw distributions of precautionary behaviors by political party.

In sum, a primary prediction made by the traditional-norms account—that social conservatism and traditionalism should correlate with pathogen avoidance—is observed, but only among Democrats, raising several questions: 1) what drives partisan differences in the relationship between socially conservative attitudes and COVID-19 precautions? and 2) as predicted by the traditional-norms account, among Democrats, are social conservatism and traditionalism better predictors of precautions than other dimensions of political attitudes?

## What drives partisan differences in the relationship between socially conservative political attitudes and COVID-19 precautions?

In order to explore what may be accounting for the observed partisan differences in the relationship between COVID-19 precautions and socially conservative attitudes, we considered the possibility that some variables—particularly those reflecting the partisan information environment dynamics and threat trade-offs discussed in the introduction—were statistically suppressing [54] an underlying relationship between precautions and socially conservative attitudes among Republicans and Independents. Specifically, though the traditional-norms account predicts an association between COVID-19 precautions and socially conservative attitudes, countervailing factors in this complex real-world context may suppress that relationship, potentially explaining the null association among Republicans and Independents reported above. Candidate variables were considered suppressors if they resulted in a significant and negative indirect pathway between socially conservative attitudes and COVID-19 precautions among Republicans and Independents. Additionally, we tested whether adjusting for suppressors would result in a) positive conditional correlations between socially conservative attitudes and COVID-19 precautions among Republicans and Independents, in contrast to the null associations at baseline, and b) non-significant interactions between socially conservative attitudes and party affiliation, such that slopes did not differ as a function of party affiliation.

In Study 1, we tested for suppression effects among Republicans and Independents across the full range of theoretically-motivated candidate variables that could plausibly be shaping partisan differences in precautionary COVID-19 behaviors, using a bottom-up exploratory approach. In order to qualify as suppression, a target variable had to have inconsistently mediated the relationship between socially conservative attitudes and precautionary behaviors among Republicans, resulting in a significant and negative indirect effect. Confidence intervals were bootstrapped for significance testing (see S3 Appendix for further details of analytic procedures, and full variable-by-variable results of the individual suppression tests).

Using this process, four variables were identified as possible suppressors among Republicans: the trust in scientists composite, the trust in liberal and moderate information sources composite, the liberal media consumption composite, and the economic conservatism composite. There was no evidence that other candidate variables were acting as suppressors, including domain-specific COVID-19 threat-assessments, and opinions about government interventions in another public health domain (smoking regulations).

In order to better visualize how these variables resulted in negative indirect effects between socially conservative attitudes and COVID-19 precautions, we regressed COVID-19 precautions on each suppressor variable, and their interactions with political party affiliation. The conditional effects were then plotted (Fig 2). In both studies, political party was a significant moderator of all four suppressor variables (see S3 Appendix for statistical details). In addition, greater trust in scientists, trust in liberals and moderates, and liberal media consumption were all positively correlated with COVID-19 precautions among Republicans and Independents. Greater economic conservatism was negatively correlated with COVID-19 precautions among

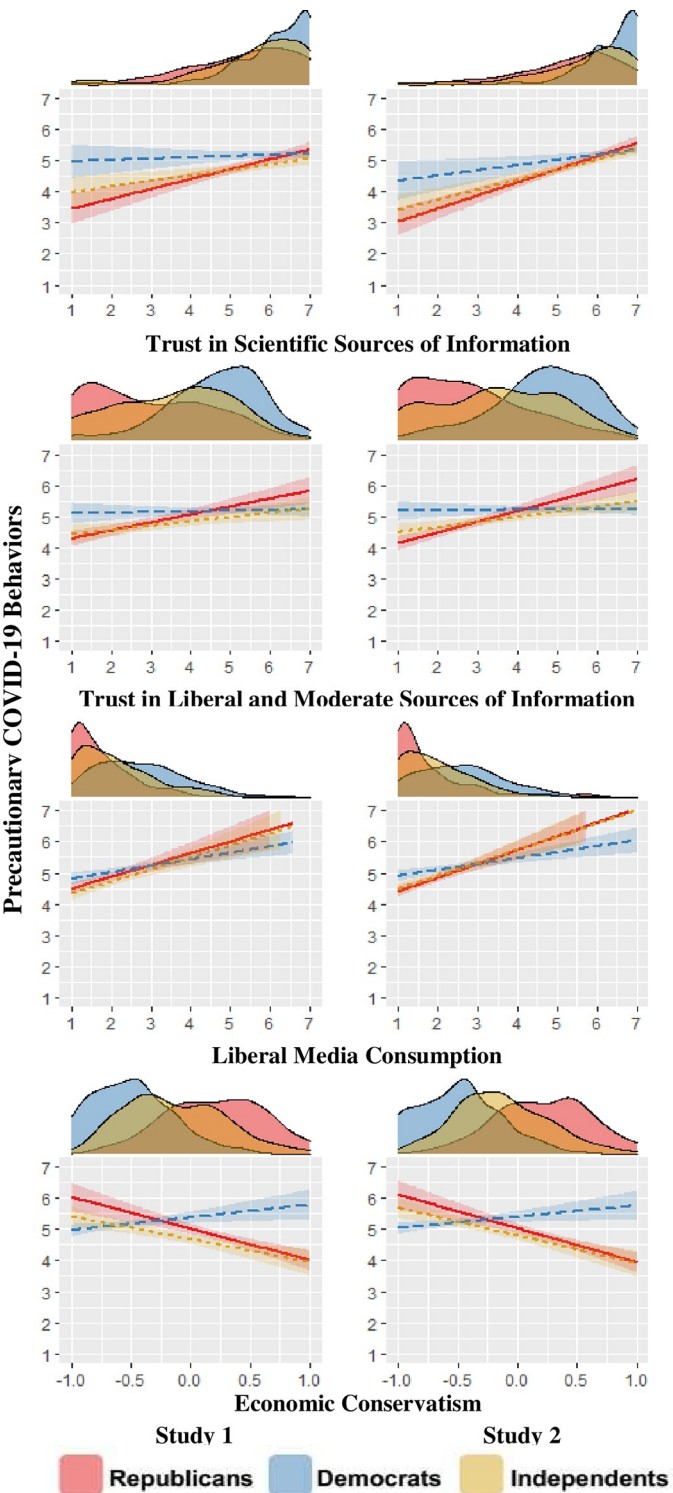

**Fig 2. Relationships between suppressor variables and COVID-19 precautions.** Studies 1 and 2 conditional effects of moderated linear regressions, in which COVID-19 precautions was regressed individually on each identified suppressor variable, political party affiliation, and the interaction between the suppressor and party affiliation. These (centered) suppressors were trust in scientists as information sources, trust in liberal and moderate figures as information sources, liberal media consumption, and economic conservatism. Bands around regression lines are 95% confidence intervals. The density plots along the x-axes represent the raw distributions of each suppressor variable by party affiliation.

Republicans and Independents (see S3 Appendix for statistical details). Further, in both Studies 1 and 2, socially conservative attitudes negatively associated with trust in scientists, trust in liberals and moderates, and economic liberalism among Republicans and Independents. Socially conservative attitudes negatively correlated with liberal media consumption among Republicans in both studies, but only among Independents in Study 2. See S3 Appendix for full details of these results.

In sum, these results illustrate the pathways by which these four variables act as suppressors of a socially conservative attitudes-precautions relationship among Republicans and Independents. First, more socially conservative attitudes were negatively correlated with greater trust in scientists and liberal and moderate sources, and greater liberal media consumption, while being positively correlated with greater economic conservatism. Second, engaging in fewer COVID-19 precautions was associated with lower trust in scientists and liberal and moderate sources of information, and lesser liberal media consumption, while being positively associated with greater economic conservatism. Taken together, these relationships result in suppression of a positive relationship between greater socially conservative attitudes and greater COVID-19 precautions among Republicans and Independents.

Because of the complex and multi-determined nature of the phenomena at hand, we considered the possibility that these four individual variables were jointly suppressing the precautions-socially conservative attitudes relationship among Republicans and Independents. Therefore, the following analyses test the combined suppressive effects of these variables.

First, we tested whether the combined effects of these four variables jointly suppressed the precautions-socially conservative attitudes relationship in Study 1. The combined indirect effect through the four candidate suppressors was negative and significant among Republicans and Independents (Republicans: bootstrapped unstandardized indirect effect = -.62, 95% CI [-.91, -.35]; Independents: indirect effect = -.43, 95% CI [-.72, -.18]), demonstrating suppression. In Study 2, we sought confirmatory evidence for the suppression model arrived at in Study 1, testing whether the combined suppressive effects of the four previously identified variables replicated, without repeating the exploratory search process of Study 1. The significant and negative indirect effect through the candidate variables replicated (Republicans: bootstrapped unstandardized indirect effect = -.40, 95% CI [-.69, -.12]; Independents: indirect effect = -.77, 95% CI [-1.06, -.50]).

Next, we further examined the effects of the suppressor variables on the relationship between socially conservative attitudes and COVID-19 precautions. We tested whether accounting for the suppressors would result in positive conditional relationships between socially conservative attitudes and precautions among Republicans and Independents. In Study 1, there was a conditional positive effect of socially conservative political attitudes on precautions among supporters of all three principal party affiliations (Fig 3); a simple slopes analysis was performed to assess those conditional effects. The simple slopes analysis indicated that, after accounting for the effects of the suppressors, the conditional effects of socially conservative attitudes were significant among Democrats ($B = .69$, SE = .17, $t(820) = 3.97$, $p < .001$), Republicans ($B = .65$, SE = .25, $t(820) = 2.64$, $p = .008$), and Independents ($B = .62$,

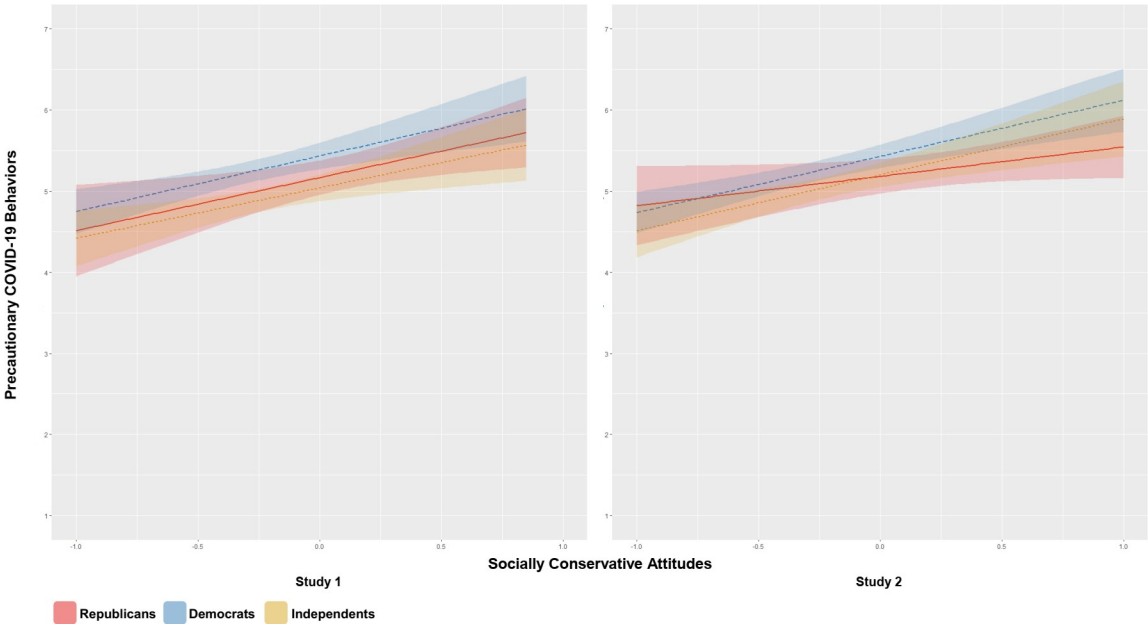

**Fig 3. Effects of suppressor variables on socially conservative attitudes–COVID-19 precautions relationship.** Studies 1 and 2 conditional effects of moderated linear regressions, in which the previously-identified suppressor variables were added to the models specified in Fig 1. These (centered) suppressors were economic conservatism, trust in scientists as information sources, trust in liberal and moderate figures as information sources, and liberal media consumption. Further, each of these suppressor variables interacted with political party affiliation in the model, because they had different effects on precautionary behavior as a function of party identification (see Fig 2 and S3 Appendix for details). Bands around regression lines are 95% confidence intervals.

SE = .20, $t(820)$ = 3.09, $p$ = .002). However, we found only partial support for these conditional relationships in Study 2: after accounting for the suppressor variables, the conditional effects were significant among Democrats ($B$ = .69, SE = .15, $t(812)$ = 4.58, $p$ = < .001) and Independents ($B$ = .69, SE = .19, $t(812)$ = 3.66, $p$ = < .001), but only approached significance among Republicans ($B$ = .36, SE = .20, $t(812)$ = 1.85, $p$ = .065). Further, after accounting for the suppressors, in both studies, slopes did not significantly differ between Democrats and Republicans (Study 1: $B$ = -.03, SE = .30, $t(820)$ = -.10, $p$ = .918; Study 2: $B$ = -.33, SE = .25, $t(812)$ = -1.34, $p$ = .182), Democrats and Independents (Study 1: $B$ = -.06, SE = .27, $t(820)$ = -.24, $p$ = .812; Study 2: $B$ = .003, SE = .24, $t(812)$ = -.01, $p$ = .991), or Republicans and Independents (Study 1: $B$ = -.03, SE = .32, $t(820)$ = -.10, $p$ = .920; Study 2: $B$ = .33, SE = .27, $t(812)$ = 1.21, $p$ = .229).

After including the suppressor variables, the party-specific socially conservative attitudes-precautions relationships were largely robust to the inclusion of basic demographic variables, as well as COVID-19-related covariates, which comprised of self-reported estimates of local COVID-19 prevalence, self-reported estimates of local population density, health status, whether participants' jobs required that they leave the home, and pathogen disgust sensitivity (see S3 Appendix). The only effect that did not obtain following inclusion of these covariates was the marginally significant conditional relationship between socially conservative attitudes and precautions among Republicans in Study 2.

## Are the relationships between socially conservative attitudes and COVID-19 precautions attributable to other dimensions of ideological attitudes?

We measured multiple dimensions of political orientation and attitude in addition to socially conservative attitudes, such as social dominance orientation and submission to authority. All

political measures were highly correlated with each other, and many also correlated with COVID-19 precautions (see S3 Appendix). Accounting for the effects of the suppressor variables, the correlations between socially conservative attitudes and COVID-19 precautions among supporters of all three major party affiliations were robust to the inclusion of the additional political ideology measures in Study 1 (see S3 Appendix). In Study 2, however, including the additional ideology variables in the moderated regressions rendered the correlation between socially conservative attitudes and COVID-19 precautions non-significant among Republicans. However, those relationships remained significant among Democrats and Independents (see S3 Appendix). Concordantly, socially conservative attitudes were the strongest positive ideological correlate of precautions among Republicans, Independents, and Democrats in Study 1, but only among Democrats and Independents in Study 2 (see S3 Appendix). Taken in sum, these results suggest that other ideological dimensions do not account for the positive socially conservative attitudes-precautions relationship, although the evidence is more consistent among Democrats and Independents relative to Republicans.

## Disgust sensitivity, politics, and precautionary COVID-19 behaviors

As noted above, pathogen disgust sensitivity did not account for the relationship between socially conservative attitudes and precautionary COVID-19 behaviors. We also tested whether pathogen disgust sensitivity was associated with COVID-19 precautionary behaviors using moderated linear regressions, where precautionary behaviors was regressed on the interaction between political affiliation and pathogen disgust sensitivity. We then performed simple slopes analyses, finding that, sensibly, in both studies, disgust sensitivity associated with precautionary behaviors among Democrats (Study 1: $B = .20$, SE = .05, $t(856) = 4.22$, $p < .001$; Study 2: $B = .24$, SE = .05, $t(854) = 4.98$, $p < .001$), Republicans (Study 1: $B = .36$, SE = .06, $t(856) = 5.63$, $p < .001$; Study 2: $B = .23$, SE = .07, $t(854) = 3.51$, $p < .001$), and Independents (Study 1: $B = .40$, SE = .06, $t(856) = 6.64$, $p < .001$; Study 2: $B = .27$, SE = .06, $t(854) = 4.41$, $p < .001$). We then used the same moderated linear regression technique to assess the relationships between pathogen disgust sensitivity and socially conservative attitudes. However, while disgust sensitivity positively correlated with socially conservative political attitudes among Democrats in both studies, as well as Independents in Study 2, there was no significant correlation among Republicans in either study, or among Independents in Study 1 (see S3 Appendix), contrary to the literature on political differences in pathogen avoidance. Because of the conceptual similarity between these results, and the party-specific effects of socially conservative attitudes on COVID-19 precautions, we tested whether economic conservatism, the trust composites, and the liberal media consumption composite were also acting as suppressors here; suppression did not account for the null association among Republicans (see S3 Appendix).

## Discussion

Partially consonant with the traditional-norms account of the relationship between political orientation and pathogen threat reactivity, in two studies, traditionalism and social conservatism correlated with COVID-19 precautionary behaviors, but the theorized relationship held only among Democrats. There was evidence that, after controlling for suppressors, these correlations appear among Republicans and Independents as well, although these findings were less robust in Study 2 than Study 1. These results are broadly consistent with previous findings that relationships between pathogen avoidance and socially conservative attitudes are stronger among liberals than conservatives [55]. We did not find support for an alternative possibility, raised in the introduction, that traditions may clash with public health COVID-19 precautions,

thus resulting in a negative correlation between socially conservative attitudes and precautions.

Simultaneously, however, the suppression of simple effects relating precaution behavior to socially conservative attitudes among Republicans and Independents indicates that clashing trade-offs between various threat domains can alter the relationships predicted by the traditional-norms account. Among Republicans and Independents, lower trust in scientists or in liberal and moderate sources, lower consumption of liberal news, as well as greater economic conservatism, appear to suppress a precautions-socially conservative attitudes association. Below, we consider possible explanations for these patterns.

Conservative politicians and news media have expressed greater doubt concerning the seriousness of the outbreak, and Republicans are less likely to trust scientists concerning COVID-19 [56]. Republicans and Democrats thus potentially occupy differing social-network and corporate-media information environments that correspond with divergent cost-mitigating responses to the pandemic. Importantly, the general relationship between media consumption and beliefs appears bidirectional. On the one hand, news content causally shapes partisan perspectives, particularly as regards beliefs about scientific issues [42, 57]. On the other hand, individuals select and trust news sources that accord with their prior views [58].

Long before the current pandemic, prominent U.S. conservatives and conservative media aggressively cast doubt on science [59], and, correspondingly, trust in science has declined among conservatives in the United States over the past four decades [60]. With regard to social conservatism, these longstanding patterns may partly owe to negative relationships between religiosity and acceptance of science [61]. With regard to economic conservatism, these patterns may partly owe to conflicts between capitalism and public-goods issues such as the societal costs of tobacco use or the shared risk of climate change.

Notably, we find that the suppressive effects of information on a positive relationship between socially conservative attitudes and prophylaxis operate not via greater trust in conservative voices, but rather via reduced trust in science, scientists, and liberal and moderate media. Because many conservative voices both question the legitimacy of scientific findings and dispute the veracity of related liberal and moderate reporting, similar considerations may apply with regard to the erosion of trust in said media, with corresponding suppressive effects on the relationship between socially conservative attitudes and COVID-19 prophylaxis.

Republicans generally report being more concerned than Democrats about the economic fallout of the pandemic [62]. We find that economic conservatism correlates with lower prophylaxis among Republicans, suggesting that more economically conservative Republicans may discount the direct health risks of the COVID-19 outbreak relative to economic considerations. Indeed, among Republicans, economic conservatism negatively correlated with concern over the direct health consequences of the pandemic (see S3 Appendix). Further, economic conservatism appears to contribute to the suppression of an underlying relationship between socially conservative attitudes and precautionary behaviors among Republicans, revealing a conflict between economic considerations and reaction to the pathogen threat posed by COVID-19. We did not find evidence that Republicans were more likely than Democrats to report taking personal steps to buffer themselves against the potential economic consequences of the pandemic (see S3 Appendix). This may be because their buffering efforts are focused on behaviors such as protesting public health orders, rather than on the economic precautions we measured; alternatively, the perceived conflict between economic considerations and responses to the health threat of COVID-19 may be primarily ideological in nature. Informational considerations may also bear on the interacting effects of economic and social conservatism in shaping prophylactic behaviors. By virtue of shared coalitional membership, the priorities of Republicans for whom economic conservatism is paramount may color the

information environment or valuations of Republicans for whom social conservatism looms largest. Addressing the concerns of powerful U.S. economic conservatives, conservative leaders and media, and other coalitional supporters in Republican social networks may prioritize the economic dangers of lockdowns and aid appropriations over the health threats of COVID-19; this may then influence social conservatives' perceptions.

These results are also consistent with the possibility of a conservative shift among more threat-sensitive Democrats, but not Republicans or Independents, in response to the pandemic. This is in contrast to previous research, which found that real-life threats in the U.S. resulted in conservative shifts among Republicans, Independents, *and* Democrats [24], findings which might have predicted a similar pattern of results across political affiliations in response to COVID-19. The suppressor variables that were observed among Republicans and Independents could plausibly be responsible for a lack of a socially conservative shift among members of those political affiliations. Specifically, as our present data show (see S3 Appendix), lower trust in scientists and liberal and moderate information sources, lower consumption of liberal media, and higher economic conservatism appear to clash with the perception that COVID-19 poses a substantial health threat. This diminished perception of hazard may weaken compensatory threat responses in social conservatives. This speaks to the importance of context-specific factors and trade-offs in structuring the relationship between threat cues and facultative shifts in socially conservative attitudes.

Regarding pathogen avoidance more broadly, we found that disgust sensitivity correlates positively with precautionary COVID-19 behaviors across individuals of all major U.S. political affiliations. Disgust reactivity is thus implicated in responses to this real-world pathogen threat, suggesting that theories such as the traditional-norms account can be justifiably applied to, and tested in relation to, this pandemic. However, socially conservative attitudes remained a robust predictor of prophylaxis after controlling for disgust sensitivity, hence precautionary behaviors aimed at avoiding COVID-19 do not appear to be fully explained by disgust responses. Measures of pathogen disgust sensitivity, including that which we employed, do not specify a context for confronting the pathogen threat, and hence the costs of avoiding the threat are unclear. In contrast, the costs of avoiding COVID-19 are real and substantial, and may be expected to vary significantly across individuals. Finally, while disgust sensitivity correlated with social conservatism among Democrats, it did not consistently do so among Republicans or Independents. Although much prior research documents a consistent and robust correlation between socially conservative attitudes and disgust sensitivity [21], our findings with regard to party affiliation are consistent with work reporting a stronger correlation among liberals than conservatives [55].

Our research is limited in important ways. First, although we found evidence that factors such as greater economic conservatism, and lower trust in scientists and liberal and moderate sources, suppressed a relationship between socially conservative attitudes and COVID-19 prophylaxis among Republicans and Independents, some of these effects did not consistently obtain. In Study 1, controlling for the combined effects of the suppressors yielded a significant positive relationship between precautionary behaviors and socially conservative attitudes among members of all major U.S. parties. However, in Study 2, accounting for the suppressors resulted in a significant relationship solely among Independents and Democrats, while that relationship only marginally approached significance among Republicans.

Second, because of the cross-sectional, correlational, non-experimental design of the research, it is impossible to draw definitive conclusions regarding causal relationships between the phenomena of interest. However, future research could in part address these limitations. For example, in regard to the relationship between socially conservative attitudes and the suppressor variables, longitudinal research might investigate the extent to which partisan media

environments shape beliefs and behaviors regarding different threats, versus the extent to which individuals seek out media environments that accord with their previous beliefs. Further, especially given the question of social conservative shifts during periods of threat, the direction of causality for the relationship between socially conservative attitudes and COVID-19 precautions found in this study is undetermined. For example, our results are consistent with the possibility of a socially conservative shift among Democrats most threatened by the pandemic; we cannot disentangle that causal pathway from one in which Democrats who were more socially conservative to begin with responded to the pandemic with greater threat avoidance. Again, longitudinal research may provide leverage on this issue. Alternatively, experimental elicitations of threat could also probe the question of directionality. However, it would be difficult to build the real-world contextualizing factors found in this study into an experimental design, thus limiting the inferential value. Finally, society-level data on the relationship between traditionalism and COVID-19 precautions may also shed light on the underlying causal relationships between the variables of interest.

Third, Republicans were slightly underrepresented in the final samples, resulting in minor power limitations when examining party-specific effects among supporters of the Republican party. Future studies should include larger samples of this group.

Fourth, we measured traditionalism using abstract questions that gauge participants' attitudes toward culture change broadly. An approach that emphasizes real behavior—the traditions that people practice, their willingness to break conventions, etc.—would offer more general validity.

Fifth, MTurk samples are not fully representative of the broader population, potentially biasing results. Likewise, the large number of surveys taken by highly-rated MTurkers such as those we employed might bias their responses. Finally, data quality can also be an issue with MTurk samples. For example, if participants are inattentive and rush through the survey in order to collect payment as quickly as possible, or if users deploy automated bots to take the survey. Despite these limitations, MTurk samples are plausibly valid for our present purposes. First, MTurk samples tend to be more diverse and attentive compared to other samples of convenience [63], and tend to replicate research conducted in population-based samples [64]. Further, and more germane to this research, psychological differences between liberals and conservatives in MTurk samples generally reflect those same differences measured in more representative samples [64]. In regard to data quality, we pre-screened for high-reputation workers—which has been shown to have a strong positive effect [65]—used re-captcha at the beginning of the survey to exclude automated bots, and included multiple attention checks. Further, attention has not been found to be worse among MTurk participants compared to participants from high-quality commercial samples, and rigorous exclusion criteria such as ours tend to increase power without compromising the sample [66].

Sixth, because population density may be a salient cue for possible exposure to SARS-CoV-2 and thus the need to engage in prophylaxis, and because Republicans and Democrats on average differ in the density of their local communities, it is possible that said differences could account for our party-specific results. However, this is unlikely given that controlling for perceived population density and COVID-19 prevalence did not account for the party-specific relationships between precautions and socially conservative attitudes.

Seventh, the dynamics studied here were only examined in the U.S., limiting the generalizability of the results. The pandemic is a global event, and questions of threat sensitivity and attitudes toward change are relevant everywhere. Indeed, although we examined political orientation in these studies—which corresponds to a particular set of social issues that are localized to a specific time and place—there are theoretical reasons to believe that traditionalism and pathogen avoidance ought to associate beyond Democrats in the U.S. political context.

Therefore, the overall generalizability of our test of the real-world validity of the traditional-norms account of the relationship between socially conservative attitudes and pathogen avoidance is limited by the lack of cross-cultural corroboration. Future work must address these same questions using cross-cultural research, particularly in nations where partisan responses to the COVID-19 outbreak have differed from those in the U.S., as well as in societies having different political and social structures, especially in regard to the value placed on traditional practices.

Our work has noteworthy strengths as well. Whereas most research on the relationship between threat sensitivity and political orientation has utilized abstract measurements that ask participants to imagine a variety of hypothetical scenarios, we asked about real behaviors in response to a widespread real-world pathogen threat, entailing actual costs and trade-offs. Socially conservative attitudes were the strongest positive predictors of precautionary behaviors relative to other dimensions of conservatism, thus our studies provide convergent real-world evidence for the traditional-norms account of the conservatism-pathogen avoidance relationship.

Further, since the start of the COVID-19 pandemic, a large body of scholarship has emerged looking at the effects of partisanship and political orientation on COVID-19 precautions and concerns in the United States [e.g. 4, 40, 67, 68]. However, many of these studies do not differentiate between political orientation and partisan identity, nor do they consider the potential interaction between them, nor examine suppressor variables linked to partisanship. These studies conclude that conservatism broadly negatively predicts COVID-19 precautions and concern. However, this result is potentially superficial, as it may owe variously to a) treating social conservatism/liberalism as isomorphic with political party affiliation; b) failure to measure distinct dimensions of ideology, such as social conservatism/liberalism, and economic conservatism/liberalism; c) failure to consider whether the effects of conservatism vary as a function of partisanship; and/or d) failure to assess suppressor variables linked to partisanship. In contrast, our results indicate that complex interactions between party affiliation and political ideology produce relationships between conservatism and COVID-19 precautions that run counter to the common assumption that conservatism is negatively associated with COVID-19 precaution. Our findings thus suggest that, both in research regarding COVID-19 and politics in the United States, and in a wide variety of related investigations, it is advisable to treat ideology and party affiliation as potentially non-substitutable, interacting variables.

Our results indicate that variation in precautionary responses to the pandemic relates to competing influences of various aspects of individuals' ideological preferences and attitudes toward change, their trust in assorted sources of information that vary along partisan dimensions, as well as the relative primacy of economic considerations. In particular, it appears that competing political factors, media consumption choices, and differences in trust may be affecting what may be underlying relationships between traditionalist social attitudes and sensitivity to pathogen threats. We speculate that Republicans—relative to Democrats—are likely exposed to and/or seek out informational environments that minimize the direct consequences of COVID-19. Instead, these informational environments may emphasize threats that resonate more strongly with the economic and libertarian dimensions of conservatism that also characterize the Republican party. These dynamics may have been amplified by the looming 2020 U.S. general election, where political and media elites in the Republican party may have been particularly motivated to downplay the threats and costs associated with the pandemic because of the potential for negative electoral consequences. As a result, an underlying relationship between socially conservative attitudes and heightened threat sensitivity may be suppressed, likely because these additional factors clash with pathogen avoidance motivations.

The present results are in tension with the current tendency to construe American partisan responses to the pandemic as defined along a simple left-right axis, where relatively liberal individuals have responded to the direct threat posed by the outbreak with greater precautions than have more conservative ones. Instead, we find that the relationship between political attitudes and reactions to the pandemic in the U.S. is complex and non-linear, such that among certain groups of individuals (i.e., Democrats) but not others (i.e., Republicans), socially conservative political attitudes are in fact associated with *greater* COVID-19 precautions—the individuals reporting taking the fewest precautions are actually more politically progressive on social issues. Lastly, we find that trust in science—and in media sources that endorse science—is associated with individual health behaviors that impact the welfare of society at large. Looking beyond the current crisis, wide variation in such trust has important implications for how the global community can best confront other worldwide threats.

## Supporting information

**S1 Appendix. Supplementary procedure.**
(PDF)

**S2 Appendix. Pilot study.**
(PDF)

**S3 Appendix. Analyses supporting main text.**
(PDF)

**S4 Appendix. Additional analyses.**
(PDF)

## Acknowledgments

We thank the UCLA Anthropology Experimental Biological Anthropology group for feedback on this research.

## Author Contributions

**Conceptualization:** Theodore Samore, Daniel M. T. Fessler, Adam Maxwell Sparks, Colin Holbrook.

**Formal analysis:** Theodore Samore, Adam Maxwell Sparks.

**Investigation:** Theodore Samore.

**Methodology:** Theodore Samore, Daniel M. T. Fessler, Adam Maxwell Sparks, Colin Holbrook.

**Project administration:** Theodore Samore.

**Visualization:** Theodore Samore.

**Writing – original draft:** Theodore Samore.

**Writing – review & editing:** Theodore Samore, Daniel M. T. Fessler, Adam Maxwell Sparks, Colin Holbrook.

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
