## [Decision Letter · Decision Letter 0]

6 Apr 2021

PONE-D-21-04873

Of Pathogens and Party Lines: Social Conservatism Positively Associates with COVID-19 Precautions among U.S. Democrats but not Republicans

PLOS ONE

Dear Dr. Samore,

Thank you for submitting your manuscript to PLOS ONE. After careful consideration, we feel that it has merit but does not fully meet PLOS ONE’s publication criteria as it currently stands. Therefore, we invite you to submit a revised version of the manuscript that addresses the points raised during the review process.

Please find below the reviewer's comments, as well as those of mine.

We look forward to receiving your revised manuscript.

Kind regards,

Valerio Capraro

Academic Editor

PLOS ONE

Additional Editor Comments:

I have now collected one review from one expert in the field. I was unable to find a second reviewer, but I myself am familiar with the topic of this manuscript, therefore I feel confident in making a decision with only one review, especially because this happens to be very detailed. The review is positive and suggests a minor revision. I tend to agree with the reviewer. Therefore, I would like to invite you to revise your work for Plos One following the reviewer's comments. Moreover, I would like to add a couple of comments mainly regarding the literature review, which I found to be incomplete. (i) the fact that there are political differences in covid-19 response has been observed in several papers (Capraro & Barcelo, 2020; Gollwitzer et al. 2020; Van Bavel et al. 2020b). This literature should be discussed. (ii) The "perspective article" on what social and behavioural science can do to support pandemic response, published by Van Bavel et al in Nature Human Behaviour, could be a useful general reference to place your paper within the literature.

I am looking forward for the revision.

Revision

Capraro, V., & Barcelo, H. (2020). The effect of messaging and gender on intentions to wear a face covering to slow down COVID-19 transmission. Journal of Behavioral Economics for Policy, 4, Special Issue 2, 45-55.

Gollwitzer, A., Martel, C., Brady, W. J., Pärnamets, P., Freedman, I. G., Knowles, E. D., & Van Bavel, J. J. (2020). Partisan differences in physical distancing are linked to health outcomes during the COVID-19 pandemic. Nature human behaviour, 4(11), 1186-1197.

Van Bavel, J. J., et al. (2020). Using social and behavioural science to support COVID-19 pandemic response. Nature Human Behaviour, 4, 460-471.

Van Bavel, J. J., et al. (2020b). National identity predicts public health support during a global pandemic. https://doi.org/10.31234/osf.io/ydt95

Journal Requirements:

[T.S., D.F., and A.S. benefited from U.S. Air Force Office of Scientific Research Award #FA9550-15-1-0137. T.S. benefited from support by the Templeton Religion Trust/Issachar Fund project “Science and Religion: An Evolutionary Perspective”.]

 [The author(s) received no specific funding for this work.]

5. We note that pictures from your S1 File "Supplementary procedures" includes images of individuals (image 3 and 7). 

Reviewers' comments:

Reviewer's Responses to Questions

**Comments to the Author**

1. Is the manuscript technically sound, and do the data support the conclusions?

Reviewer #1: Partly

2. Has the statistical analysis been performed appropriately and rigorously? 

Reviewer #1: Yes

3. Have the authors made all data underlying the findings in their manuscript fully available?

Reviewer #1: Yes

4. Is the manuscript presented in an intelligible fashion and written in standard English?

Reviewer #1: Yes

5. Review Comments to the Author

Reviewer #1: Of Pathogens and Party Lines: Social Conservatism Positively Associates with COVID-19 Precautions among U.S. Democrats but not Republicans

PLOS ONE

The paper reports two correlational M-Turk studies with samples of U.S. American adults and investigates potential ideological differences in COVID-19 threat reactivity. Specifically, the research investigated the relationship between social conservatism and COVID-19 precautionary behavior in light of the hyper-politicized partisan landscape of the United States.

In accordance with a model wherein ideological differences stem from differences in threat reactivity, they found that social conservatives (only among democrats) tended to be more pathogen-avoidant (based on the degrees of COVID-19 precautionary behavior). However, this was not found to be true for either Independents or Republicans. Their approach, both theoretical and empirical, to provide an explanation to why this was not the case for the most conservative partisan group (Republicans), was in my opinion satisfactory and yield a good theoretical and empirical account.

As a whole I believe that when put in more context with the literature on threat-sensitivity and political ideology, which I recommend doing so, this manuscript could also contribute to the question of a lack of conservative shift during the COVID-19 crisis in the United States. However, there are some issues that should be addressed before suggesting this manuscript for publication. Issues concerning the general contextualization, the body of literature covered, and more crucially, the reporting of the study’s results should be significantly addressed before recommending this manuscript for publication. In what follows I include some areas to consider for improvement.

-In general, I felt that the introduction to the manuscript (p. 3, line 49) could benefit from taking a step back from the theory and the data-driven points. It would be good, before going into the theory, to provide a general description of the political state of affairs and dynamics in the U.S. with relation to the COVID-19 pandemic at the time of the project/data collection. Following from the general rationale of the first sentence of the introduction, I feel that a more detailed description of the situation (the hyper-politization of the pandemic) would facilitate a better segue from the real-world problem to the theoretical research question. Perhaps a couple more sentences (or maybe an additional paragraph) that will situate the reader in the very real and specific time and moment of the COVID-19 Pandemic and the general political landscape (Election Year in the U.S.), will provide further justification to the timely question this manuscript addresses.

-On another but related point, the question of conservative shift (e.g., Bonanno & Jost, 2006; Beall, Hofer, & Schaller, 2016) under real-life common threats in the case of COVID-19 seems to be one important question that the paper does not addresses directly, but that would be important to do so. Maybe addressing this at the beginning, or at some point in the introduction would also put the results and findings in conversation with this general theoretical premise from a threat-sensitivity (or system-threat) perspective. Based on the threat literature and political ideology, some might have predicted potential political ideological shifts (a conservative one) based on exposure to the real life (and the multiple) threats that COVID poses. It seems like they find this pattern to be present only among Democrats (which is interesting in its own way), and that other political factors suppressed this in the case of Republicans.

I wonder if contextualizing the findings with respect to the suppressor variables (e.g., lower trust in scientists, liberal and moderate sources, lesser consumption of liberal news media, and greater economic conservatism) could further help clarify why, at least in this case, a general move from social conservatism toward increases in protective behaviors against COVID-19 was not observed. The incorporation of the question of ideological shifts, as well as the real-life implication of the impact of COVID-19 on the 2020 elections (either in the introduction or general discussion), will strengthen the theoretical contribution of this manuscript.

-(p. 12, line 264) When describing the data collection periods, the specific dates for the data collection period should be included when describing the pilot study as well. It may be useful as well, based on my impression of the supplementary materials, to include time effect differences on the correlations of social conservatism and COVID-19 precautionary behavior, given that they found no significant difference between Republicans and Democrats in their initial pilot study:

“We find no significant baseline differences in mean precautionary behaviors between Republicans (M = 5.05, SD = 1.31) and Democrats (M = 5.20, SD = 1.17) in our sample (t[215.68] = -1.08, p = .282).” (p. 9, supplement).

It could very well be, that Time might be playing a role with regards the strength and significance of this relationship. Therefore, it might be appropriate to examine if this positive association keeps fading away with time, as the polarization and politization of COVID-19 precautionary mandates in the United States kept increasing with time. This could go in a footnote, or even in one of the supplements.

- (pp. 19-25) Perhaps my major concern about the current state of the manuscript has to do with the general Results section. I recommend substantial rewriting of this section. As it stands right now, the present iteration of the first section subtitled “Do socially conservative political attitudes predict precautionary behavior?” seems vague and does not follow a strong line of organization. For instance, the first paragraph claims that “using linear regression with moderation, in both studies, COVID-19 prophylaxis associated with socially conservative political attitudes among Democrats” (p. 19, line 431)” but they do not provide any statistics to support this claim in the main text. I feel that the authors did a good job of reporting this results in their supplement, but they seem to have excluded important statistical information in the main text. I suggest integrating certain components from their supplementary material into the main text, because there is just too little information provided in the result section of the main text. Furthermore, with regards to this specific section, the description of the figure should not be in the text body, but rather as a note for the figure. I am not sure if they intended to include this information here, of if it was meant to be part of the description of the figure.

- For instance, I recommend that the part on p. 20 (lines 400-451) be used as the note for the figure, and the rest about simple slope analyses should be integrated into the main text. The same is true for the results reported on p. 22 (line 480). Because these all seems issues of style and formatting, I am confident that the authors will be able to address these concerns.

In the supplement the authors demonstrate the step-by-step analytical process to determine the general suppressors in a very satisfactory way. I would recommend they also integrate some of this detailed analysis in the main text, because it seems very important (p. 9, supplement).

Finally, I would also recommend including the analyses of conditional effects of moderated linear regressions in which COVID-19 precautions were separately regressed on each individual (centered) suppressor variable for Study 1 and Study 2 (p. 15, supplement). This analysis really makes the argument and findings clearer and more telling. In summary, then, I believe that a general reworking of the Results section is advisable prior to publication.

MINOR CONCERNS:

-(p. 5, line 110) The authors wrote:

“Thus, prior research on the traditional-norms account has not sufficiently taken into account the cost-benefit trade-offs of many threat and pathogen avoidance behaviors, in turn limiting the ecological validity of the findings.”

This is a turning point in the introduction that leads to one of the central premises of their hypotheses, and it highlights the magnitude of the present contribution. As it stands right now, the text seems a bit vague and overly general. To enhance the strength of their argument, I would recommend the authors directly cite the work they are alluding to here and incorporate any other studies that have already tried to address those limitations (successfully or not). I feel the paper would benefit from being as direct and concise as possible with regards to the work they are citing, constructively criticizing, and building upon.

-(p. 6, line 128-138) The authors wrote:

“Some of the precautions recommended or required by public health authorities interfere with engaging in traditional practices (e.g., social distancing precludes family gatherings, public sporting events, in-person religious services, etc.). Accordingly, two possibilities exist regarding the relationship between threat sensitivity, socially conservative attitudes, and COVID-19 precautions. On the one hand, if more threat-sensitive individuals focus on the precautions themselves, construing these as infringing on traditions, then they will report lower precautionary behavior than less threat-sensitive individuals–and, indeed, may view such behaviors as threats in themselves, endangering individual liberties. On the other hand, if more threat-sensitive individuals focus on the danger posed by COVID-19 more so than the conflict with various traditions entailed by precautionary behaviors, then they will report both greater precautionary behavior and greater valuation of traditions than will less threat-sensitive individuals.”

I really like this conceptualization, and it helps facilitate a more holistic understanding of the dynamics of social conservatism and threat-avoidance in a more complex way than one normally encounters (which is a simple linear, “cause and effect” way). One recommendation, however, would be to elaborate on the explanation of the first possibility, because it is more novel in political psychology. I recommend describing the second possibility first (because it is the simpler one), and then discussing the one about “focusing on the precautions themselves.” I recommend turning the 4-line sentence, into perhaps two shorter sentences for the purpose of clarity.

-(p. 11, line 241) typo: “In” after last sentence of section.

-(p. 14, line 301) All the measures included here should be presented in the precise order that they were presented to participants in the study. Also, is the order of measures consistent across studies?

-(p. 14, line 320) I would recommend elaborating further on the development of the social conservative attitude scale, because it is the central variable of interest and the operationalization deviates somewhat from the more “traditional” or “common” conception and usage of ideology (the left-right ideological self-placement item). I suggest that the authors devote more space explaining and providing concrete examples about the development and operationalization of this variable in the text of the manuscript.

-(p. 16, line 358) It would be useful if there were sample items provided, rather than a description of the contents of these items.

-(p. 16, line 360) I would recommend using sub-headers for each of the 6 categories, to make the last section easier to follow. As it reads now, this passage may lead to a bit of confusion. For example, the authors write:

(p. 17, line 382):

“Third, participants were asked about the extent to which they were preparing for an economic downturn, such as by delaying major financial decisions. We averaged these behavior items into an economic precautions composite, which was reliable (αs = .75 – .78).”

I will recommend doing something like this:

3. Economic precautions: Participants were asked about the extent to which they were preparing for an economic downturn by delaying major financial decisions. We averaged these behavioral items into a composite scale, which was reliable (αs = .75 – .78).

-(p. 19, line 416) It would be useful to remind the reader of the total sample size and report the precise number of self-identified “Other” parties.

-(p. 19, line 423 and throughout the results section) Many of the reported statistics (e.g., p = 3.33e-6) do not conform to APA style.

-(pp. 20, line 452) They report:

"Finally, in both studies, the interactions between political party affiliation and socially conservative attitudes were significant between Democrats on the one hand, and both Republicans (Study 1: B = -0.80, SE = .29, t(857) = -2.77, p = .006; Study 2: B = -0.79, SE = .26, t(854) = -3.04, p = .002) and Independents (Study 1: B = -0.69, SE = .25, t(857) = -2.72, p = .007; Study 2: B = -0.78, SE = .23, t(854) = -3.39, p = .001) on the other. Slopes did not significantly differ between Independents and Republicans (Study 1: B = 0.11, SE = .31, t(857) = .35, p = .728; Study 2: B = 0.01, SE = .28, t(854) = .04, p = .969)."

It is not clear what are they trying to say with this. It seems like the only thing they should report here is that the slopes between Independents and Republicans did not differ from each other. The same is true on p. 22 (line 499). I recommend clarifying these points.

-(p. 24, line 547) The authors reported combined statistics like this:

"We then performed simple slopes analyses, finding that, sensibly, in both studies, disgust sensitivity associated with precautionary behaviors among Democrats (Bs = .20 – .24, ps = 7.60e-7 – 2.66e-5), Republicans (Bs = .23  – .36, ps = 2.40e-8 – 4.76e-4), and Independents (Bs = .27 – .40, ps = 5.63e-11 – 1.16e-5)."

I would recommend reporting these statistics separately for each study in a consistent way throughout the Results section.

-(p. 29, line 648-651) They wrote:

“[…] Nevertheless, in both studies, we found significant evidence that suppression by the target variables occurred, such that, when accounting for the suppressors, political party no longer moderated the relationship between socially conservative attitudes and precautions.”

The last sentence in the first limitations paragraph seems a bit contradictory to the main points they are raising just before. I recommend to simply take it out.

-(p. 29, line 652) They wrote:

“Second, because of the cross-sectional, correlational, non-experimental design of the research, it is impossible to draw definitive conclusions regarding causal relationships between the phenomena of interest.”

This statement seems pretty vague, and it would be beneficial to suggest ways of overcoming this limitation. Perhaps elaborate a bit more on the specific causal relationship you would like to address and report ways that one could potentially do so with different types of data or research designs.

Concluding remarks:

In sum, the inconsistencies with regards to the organization and content of the results section made the paper somewhat difficult to follow and comprehend. It was difficult to get a clear picture from the Results section in the main text alone, and without the supplemental material it was difficult to get a full picture of the results, as well as the analytical strategies implemented. I strongly recommend that these issues are resolved prior to publication.

6. PLOS authors have the option to publish the peer review history of their article (what does this mean?). If published, this will include your full peer review and any attached files.

Reviewer #1: No

---

## [Author Response · Author response to Decision Letter 0]

26 May 2021

Dear Dr. Capraro,

We thank you and the Reviewer for the careful assessment of our submission (PONE-D-21-04873). Below, in italics, we address each of your comments, and those of the Reviewer, in turn. To enhance clarity, we have numbered these comments in order to facilitate cross-referencing. 

We look forward to your assessment.

Sincerely,

Theodore Samore (on behalf of all authors)

EDITOR’S PRINCIPAL REMARKS:

E1 Moreover, I would like to add a couple of comments mainly regarding the literature review, which I found to be incomplete. (i) the fact that there are political differences in covid-19 response has been observed in several papers (Capraro & Barcelo, 2020; Gollwitzer et al. 2020; Van Bavel et al. 2020b). This literature should be discussed. (ii) The "perspective article" on what social and behavioural science can do to support pandemic response, published by Van Bavel et al in Nature Human Behaviour, could be a useful general reference to place your paper within the literature.

REPLY: We thank the Editor for bringing these citations to our attention; we agree that they are important additions to the literature review in our manuscript. We have therefore added context to the Introduction section by including this literature, see lines 201-216.

E2 Please ensure that your manuscript meets PLOS ONE's style requirements, including those for file naming. 

Reply: We have edited the manuscript to ensure that it meets PLOS ONE’s style requirements

E3 Please review your reference list to ensure that it is complete and correct. If you have cited papers that have been retracted, please include the rationale for doing so in the manuscript text, or remove these references and replace them with relevant current references. Any changes to the reference list should be mentioned in the rebuttal letter that accompanies your revised manuscript. If you need to cite a retracted article, indicate the article’s retracted status in the References list and also include a citation and full reference for the retraction notice.

REPLY: Thank you, we have reviewed our reference list to ensure that it is complete and correct. We have also added additional citations throughout the manuscript to more thoroughly contextualize different claims. All changes to the references have been highlighted with track changes in the ‘Revised Manuscript with Track Changes’ document.

E4 Thank you for stating the following in the Acknowledgments Section of your manuscript:

[T.S., D.F., and A.S. benefited from U.S. Air Force Office of Scientific Research Award #FA9550-15-1-0137. T.S. benefited from support by the Templeton Religion Trust/Issachar Fund project “Science and Religion: An Evolutionary Perspective”.]

 [The author(s) received no specific funding for this work.]

REPLY: Thank you for bringing this error to our attention. We have changed the Acknowledgments Section of the manuscript to remove all funding information (see lines 973-975). Further, we would like to update our Funding Statement with the following text:

T.S., D.F., and A.S. benefited from U.S. Air Force Office of Scientific Research Award #FA9550-15-1-0137. T.S. benefited from support by the Templeton Religion Trust/Issachar Fund project “Science and Religion: An Evolutionary Perspective”.

E5 Please include captions for your Supporting Information files at the end of your manuscript, and update any in-text citations to match accordingly. Please see our Supporting Information guidelines for more information: http://journals.plos.org/plosone/s/supporting-information.

REPLY: We have added captions for the Supporting Information files at the end of the manuscript (see lines 1195-1199), and we have updated in-text citations accordingly.

E6 We note that pictures from your S1 File "Supplementary procedures" includes images of individuals (image 3 and 7). 

REPLY: We have removed all images that may contain images of individuals in the Supplementary Procedures file.

REVIEWER’S COMMENTS TO AUTHOR:

R1 The manuscript must describe a technically sound piece of scientific research with data that supports the conclusions. Experiments must have been conducted rigorously, with appropriate controls, replication, and sample sizes. The conclusions must be drawn appropriately based on the data presented.

Reviewer #1: Partly

REPLY: We thank the Reviewer for their thorough and insightful review of the manuscript. We feel that our responses to the Reviewer’s feedback should assuage any concerns that our conclusions are only partly supported by the data. 

R2 In general, I felt that the introduction to the manuscript (p. 3, line 49) could benefit from taking a step back from the theory and the data-driven points. It would be good, before going into the theory, to provide a general description of the political state of affairs and dynamics in the U.S. with relation to the COVID-19 pandemic at the time of the project/data collection. Following from the general rationale of the first sentence of the introduction, I feel that a more detailed description of the situation (the hyper-politization of the pandemic) would facilitate a better segue from the real-world problem to the theoretical research question. Perhaps a couple more sentences (or maybe an additional paragraph) that will situate the reader in the very real and specific time and moment of the COVID-19 Pandemic and the general political landscape (Election Year in the U.S.), will provide further justification to the timely question this manuscript addresses.

REPLY: We agree with the Reviewer that situating this research within the broader context of the COVID-19 pandemic in the U.S. at the time of data collection will make the theoretical contributions of the work clearer. Accordingly, we have added the following prose to the beginning of the Introduction section (lines 47-77):

In the spring and summer of 2020, the COVID-19 pandemic was profoundly shaping the personal, social, and political lives of most Americans. Although case counts across most of the United States were relatively low compared to the subsequent fall and winter waves, the pandemic’s effects were already widely felt. Many people adopted a suite of prophylactic behaviors, including mask wearing, social distancing, disinfecting, and social isolation to avoid infection. Many businesses, services, and schools were ordered closed in order to stem the spread of the pandemic. In turn, the effects of the pandemic and subsequent closures resulted in substantial economic decline, sparking political debate about both the cost-benefit trade-offs of COVID-19-related restrictions, as well as the nature and extent of economic relief measures. Notably, the pandemic was also heavily politicized (Hart et al., 2020). In general, politicians from the Republican party and sympathetic media figures downplayed the direct health severity of the pandemic relative to their Democratic counterparts, while emphasizing the costs of closures and restrictions to both the economy and personal liberty. 

Potentially motivated by the lead-up to a presidential election in November 2020 and a desire to minimize a national crisis that could negatively impact his electability, then-President Donald Trump and his allies in the Republican party consistently downplayed the threat posed by the pandemic, claiming variably that the virus would disappear, that it was not any more dangerous than seasonal flu, and that prophylactic measures such as mask wearing were unnecessary (Paz, 2020). Polling and research suggests that these attitudes among party elites were also reflected among supporters of the Republican party (Pew Research Center, 2020c), including in their own health-related behaviors such as social distancing and mask-wearing (Gadarian et al., 2021; Gollwitzer et al., 2020). 

Yet, in contrast to this dynamic in the United States where Republicans – the more socially conservative party – have been more skeptical than Democrats of the dangers of the COVID-19 pandemic, a large previous literature has both theorized and demonstrated a positive association between social conservatism and sensitivity to threats, particularly threats from pathogens (see Hibbing et al, 2014). Here, in two studies conducted in the spring and summer of 2020, we explore the partisan patterns of U.S. responses to COVID-19 as a case study that challenges theoretical frameworks that link together political orientation, attitudes toward traditional norms, threat sensitivity, partisanship, and cost-benefit trade-off calculations between competing sources of threat.

R3 On another but related point, the question of conservative shift (e.g., Bonanno & Jost, 2006; Beall, Hofer, & Schaller, 2016) under real-life common threats in the case of COVID-19 seems to be one important question that the paper does not addresses directly, but that would be important to do so. Maybe addressing this at the beginning, or at some point in the introduction would also put the results and findings in conversation with this general theoretical premise from a threat-sensitivity (or system-threat) perspective. Based on the threat literature and political ideology, some might have predicted potential political ideological shifts (a conservative one) based on exposure to the real life (and the multiple) threats that COVID poses. It seems like they find this pattern to be present only among Democrats (which is interesting in its own way), and that other political factors suppressed this in the case of Republicans. 

I wonder if contextualizing the findings with respect to the suppressor variables (e.g., lower trust in scientists, liberal and moderate sources, lesser consumption of liberal news media, and greater economic conservatism) could further help clarify why, at least in this case, a general move from social conservatism toward increases in protective behaviors against COVID-19 was not observed. The incorporation of the question of ideological shifts, as well as the real-life implication of the impact of COVID-19 on the 2020 elections (either in the introduction or general discussion), will strengthen the theoretical contribution of this manuscript.

REPLY: We welcome the suggestion of addressing the question of conservative shifts in response to real-life threats, as that literature ties in closely to the issues and theory being addressed in this paper. Therefore, we have added the following paragraph to the Introduction that discusses conservative shifts, and attends particularly to how such shifts dovetail with the theory presented earlier in the manuscript (lines 150-159). 

Concordantly, a related body of scholarship has theorized and empirically tested the possibility of conservative shifts in response to real-life threats (Beall et al., 2016; Bonanno & Jost, 2006; Claessens et al., 2020; Nail & Mcgregor, 2009; but see Tiokhin & Hruschka, 2016). The underlying functional logic is shared with the traditional norms account: if traditions and socially conservative norms can mitigate the costs of recurrent threats, then cues of increased threat may lead individuals to flexibly upregulate their traditionalism and social conservatism in response. Indeed, the possibility that temporal variation in threats results in conservative shifts is not mutually exclusive with the possibility that trait threat sensitivity influences social conservatism. Rather, both relationships may result from a shared underlying process that links threat to resistance to social change. 

Further, we have added additional contextualization and discussion of conservative shifts throughout the manuscript. First, we mention how the literature on conservative shifts relates to the question of threat trade-offs and ecological validity (lines 170-172). Second, we cite a growing literature on the question of a conservative shift in response to the COVID-19 pandemic (lines 201-205). 

Relatedly, the Reviewer also commented as follows:

“As a whole I believe that when put in more context with the literature on threat-sensitivity and political ideology, which I recommend doing so, this manuscript could also contribute to the question of a lack of conservative shift during the COVID-19 crisis in the United States.”

We have therefore added prose to the Discussion section detailing how our results are consistent with a conservative shift among Democrats, but not among Republicans and Independents, contrary to prior findings. We then contextualize how the suppressor variables may have interfered with such a conservative shift among Republicans (lines 817-830). 

These results are also consistent with the possibility of a conservative shift among more threat-sensitive Democrats, but not Republicans or Independents, in response to the pandemic. This is in contrast to previous research, which found that real-life threats in the U.S. resulted in conservative shifts among Republicans, Independents, and Democrats (Bonanno & Jost, 2006), findings which might have predicted a similar pattern of results across political affiliations in response to COVID-19. The suppressor variables that were observed among Republicans and Independents could plausibly be responsible for a lack of a socially conservative shift among members of those political affiliations. Specifically, as our present data show (see S3 Appendix), lower trust in scientists and liberal and moderate information sources, lower consumption of liberal media, and higher economic conservatism appear to clash with the perception that COVID-19 poses a substantial health threat. This diminished perception of hazard may weaken compensatory threat responses in social conservatives. This speaks to the importance of context-specific factors and trade-offs in structuring the relationship between threat cues and facultative shifts in socially conservative attitudes. 

Finally, in regard to the implications of COVID-19 and our results vis a vis the 2020 elections, we added prose to the Introduction and Discussion sections considering the possibility that concerns over the election may have in part shaped elite messaging that subsequently contributed to the suppression of a social conservatism-precautions relationship (lines 60-65, and lines 954-956). 

R4 When describing the data collection periods, the specific dates for the data collection period should be included when describing the pilot study as well. It may be useful as well, based on my impression of the supplementary materials, to include time effect differences on the correlations of social conservatism and COVID-19 precautionary behavior, given that they found no significant difference between Republicans and Democrats in their initial pilot study: 

“We find no significant baseline differences in mean precautionary behaviors between Republicans (M = 5.05, SD = 1.31) and Democrats (M = 5.20, SD = 1.17) in our sample (t[215.68] = -1.08, p = .282).” (p. 9, supplement). 

It could very well be, that Time might be playing a role with regards the strength and significance of this relationship. Therefore, it might be appropriate to examine if this positive association keeps fading away with time, as the polarization and politization of COVID-19 precautionary mandates in the United States kept increasing with time. This could go in a footnote, or even in one of the supplements. 

REPLY: We agree that it is important to include the date of the pilot study as well, in order to contextualize each study in a particular time, especially given how quickly the situation around the pandemic was changing. Line 336 was changed to include that date.

In regard to the issue of whether time may be playing a role with regard to the relationship between political party affiliation and COVID-19 precautions—and in particular, the lack of a significant difference between Republicans and Democrats in the pilot study relative to the differences that obtained in Studies 1 and 2—we agree that it’s very plausible that increasing politicization and polarization over the course of the pandemic (and over the course of the three studies presented here) may have led to starker differences in precautions between Democrats and Republicans. To test this possibility, we ran a two-way ANOVA that examined the effects of time (measured across the pilot study, and Studies 1 and 2) and political party affiliation on COVID-19 precautions. Despite the plausibility of the hypothesis, we found neither an effect of time on precautions among either Republicans or Democrats, nor a significant interaction between time and party affiliation. However, given that this null result is in itself interesting, we report it in S4 Appendix (pages 50-52). 

R5 Perhaps my major concern about the current state of the manuscript has to do with the general Results section. I recommend substantial rewriting of this section. As it stands right now, the present iteration of the first section subtitled “Do socially conservative political attitudes predict precautionary behavior?” seems vague and does not follow a strong line of organization. For instance, the first paragraph claims that “using linear regression with moderation, in both studies, COVID-19 prophylaxis associated with socially conservative political attitudes among Democrats” (p. 19, line 431)” but they do not provide any statistics to support this claim in the main text. I feel that the authors did a good job of reporting this results in their supplement, but they seem to have excluded important statistical information in the main text. I suggest integrating certain components from their supplementary material into the main text, because there is just too little information provided in the result section of the main text. Furthermore, with regards to this specific section, the description of the figure should not be in the text body, but rather as a note for the figure. I am not sure if they intended to include this information here, of if it was meant to be part of the description of the figure. 

For instance, I recommend that the part on p. 20 (lines 400-451) be used as the note for the figure, and the rest about simple slope analyses should be integrated into the main text. The same is true for the results reported on p. 22 (line 480). Because these all seems issues of style and formatting, I am confident that the authors will be able to address these concerns. 

REPLY: We agree that the Results section was insufficiently clear and comprehensive. Therefore, we substantially modified and expanded the Results section to address the Reviewer’s concerns. First, we moved the statistics presented in the caption of Figure 1 to the main text, under the header, “Do socially conservative attitudes predict precautionary behavior?” (see lines 560-569), which should address the Reviewer’s concern about the lack of statistics used to support the claims in this section. Likewise, we similarly moved the results from line 480 in the original manuscript to the main text (see lines 673-689 in the updated manuscript). Second, we have made it more clear that the description of Figure 1, as well as the descriptions of other figures in the main text, are captions for those figures, rather than parts of the main text body (see lines 574-581, lines 636-643, and lines 691-699).

R6 In the supplement the authors demonstrate the step-by-step analytical process to determine the general suppressors in a very satisfactory way. I would recommend they also integrate some of this detailed analysis in the main text, because it seems very important (p. 9, supplement). 

REPLY: We appreciate this suggestion, and agree that it strengthens the manuscript to include a more thorough description of the suppression-related analyses in the main text. Therefore, we have substantially added to the section titled “What drives partisan differences in the relationship between socially conservative political attitudes and COVID-19 precautions”, including the addition of material that was presented in Supplement S3 in the previous submission; see below (lines 589-708).

In order to explore what may be accounting for the observed partisan differences in the relationship between COVID-19 precautions and socially conservative attitudes, we considered the possibility that some variables—particularly those reflecting the partisan information environment dynamics and threat trade-offs discussed in the introduction—were statistically suppressing (MacKinnon et al., 2000) an underlying relationship between precautions and socially conservative attitudes among Republicans and Independents. Specifically, though the traditional-norms account predicts an association between COVID-19 precautions and socially conservative attitudes, countervailing factors in this complex real-world context may suppress that relationship, potentially explaining the null association among Republicans and Independents reported above. Candidate variables were considered suppressors if they resulted in a significant and negative indirect pathway between socially conservative attitudes and COVID-19 precautions among Republicans and Independents. Additionally, we tested whether adjusting for suppressors would result in a) positive conditional correlations between socially conservative attitudes and COVID-19 precautions among Republicans and Independents, in contrast to the null associations at baseline, and b) non-significant interactions between socially conservative attitudes and party affiliation, such that slopes did not differ as a function of party affiliation.

In Study 1, we tested for suppression effects among Republicans and Independents across the full range of theoretically-motivated candidate variables that could plausibly be shaping partisan differences in precautionary COVID-19 behaviors, using a bottom-up exploratory approach. In order to qualify as suppression, a target variable had to have inconsistently mediated the relationship between socially conservative attitudes and precautionary behaviors among Republicans, resulting in a significant and negative indirect effect. Confidence intervals were bootstrapped for significance testing (see S3 Appendix for further details of analytic procedures, and full variable-by-variable results of the individual suppression tests). 

Using this process, four variables were identified as possible suppressors among Republicans: the trust in scientists composite, the trust in liberal and moderate information sources composite, the liberal media consumption composite, and the economic conservatism composite. There was no evidence that other candidate variables were acting as suppressors, including domain-specific COVID-19 threat-assessments, and opinions about government interventions in another public health domain (smoking regulations). 

In order to better visualize how these variables resulted in negative indirect effects between socially conservative attitudes and COVID-19 precautions, we regressed COVID-19 precautions on each suppressor variable, and their interactions with political party affiliation. The conditional effects were then plotted (Fig 2). In both studies, political party was a significant moderator of all four suppressor variables (see S3 Appendix for statistical details). In addition, greater trust in scientists, trust in liberals and moderates, and liberal media consumption were all positively correlated with COVID-19 precautions among Republicans and Independents. Greater economic conservatism was negatively correlated with COVID-19 precautions among Republicans and Independents (see S3 Appendix for statistical details). Further, in both Studies 1 and 2, socially conservative attitudes negatively associated with trust in scientists, trust in liberals and moderates, and economic liberalism among Republicans and Independents. Socially conservative attitudes negatively correlated with liberal media consumption among Republicans in both studies, but only among Independents in Study 2. See S3 Appendix for full details of these results. 

In sum, these results illustrate the pathways by which these four variables act as suppressors of a socially conservative attitudes-precautions relationship among Republicans and Independents. First, more socially conservative attitudes were negatively correlated with greater trust in scientists and liberal and moderate sources, and greater liberal media consumption, while being positively correlated with greater economic conservatism. Second, engaging in fewer COVID-19 precautions was associated with lower trust in scientists and liberal and moderate sources of information, and lesser liberal media consumption, while being positively associated with greater economic conservatism. Taken together, these relationships result in suppression of a positive relationship between greater socially conservative attitudes and greater COVID-19 precautions among Republicans and Independents.

Because of the complex and multi-determined nature of the phenomena at hand, we considered the possibility that these four individual variables were jointly suppressing the precautions-socially conservative attitudes relationship among Republicans and Independents. Therefore, the following analyses test the combined suppressive effects of these variables.

First, we tested whether the combined effects of these four variables jointly suppressed the precautions-socially conservative attitudes relationship in Study 1. The combined indirect effect through the four candidate suppressors was negative and significant among Republicans and Independents (Republicans: bootstrapped unstandardized indirect effect = -.62, 95% CI [-.91, -.35]; Independents: indirect effect = -.43, 95% CI [-.72, -.18]), demonstrating suppression. In Study 2, we sought confirmatory evidence for the suppression model arrived at in Study 1, testing whether the combined suppressive effects of the four previously identified variables replicated, without repeating the exploratory search process of Study 1. The significant and negative indirect effect through the candidate variables replicated (Republicans: bootstrapped unstandardized indirect effect = -.40, 95% CI [-.69, -.12]; Independents: indirect effect = -.77, 95% CI [-1.06, -.50]).

Next, we further examined the effects of the suppressor variables on the relationship between socially conservative attitudes and COVID-19 precautions. We tested whether accounting for the suppressors would result in positive conditional relationships between socially conservative attitudes and precautions among Republicans and Independents. In Study 1, there was a conditional positive effect of socially conservative political attitudes on precautions among supporters of all three principal party affiliations (Fig 2); a simple slopes analysis was performed to assess those conditional effects. The simple slopes analysis indicated that, after accounting for the effects of the suppressors, the conditional effects of socially conservative attitudes were significant among Democrats (B = .69, SE = .17, t(820) = 3.97, p < .001), Republicans (B = .65, SE = .25, t(820) = 2.64, p = .008), and Independents (B = .62, SE = .20, t(820) = 3.09, p = .002). However, we found only partial support for these conditional relationships in Study 2: after accounting for the suppressor variables, the conditional effects were significant among Democrats (B = .69, SE = .15, t(812) = 4.58, p = < .001) and Independents (B = .69, SE = .19, t(812) = 3.66, p = < .001), but only approached significance among Republicans (B = .36, SE = .20, t(812) = 1.85, p = .065). Further, after accounting for the suppressors, in both studies, slopes did not significantly differ between Democrats and Republicans (Study 1: B = -.03, SE = .30, t(820) = -.10, p = .918; Study 2: B = -.33, SE = .25, t(812) = -1.34, p = .182), Democrats and Independents (Study 1: B = -.06, SE = .27, t(820) = -.24, p = .812; Study 2: B = .003, SE = .24, t(812) = -.01, p = .991), or Republicans and Independents (Study 1: B = -.03, SE = .32, t(820) = -.10, p = .920; Study 2: B = .33, SE = .27, t(812) = 1.21, p = .229).

After including the suppressor variables, the party-specific socially conservative attitudes-precautions relationships were largely robust to the inclusion of basic demographic variables, as well as COVID-19-related covariates, which comprised of self-reported estimates of local COVID-19 prevalence, self-reported estimates of local population density, health status, whether participants’ jobs required that they leave the home, and pathogen disgust sensitivity (see S3 Appendix). The only effect that did not obtain following inclusion of these covariates was the marginally significant conditional relationship between socially conservative attitudes and precautions among Republicans in Study 2. 

R7 Finally, I would also recommend including the analyses of conditional effects of moderated linear regressions in which COVID-19 precautions were separately regressed on each individual (centered) suppressor variable for Study 1 and Study 2 (p. 15, supplement). This analysis really makes the argument and findings clearer and more telling. In summary, then, I believe that a general reworking of the Results section is advisable prior to publication. 

REPLY: We agree that including this analysis in the main text makes the argument of this section clearer. We have therefore added an additional figure (Figure 2) to the main text, which shows the conditional effects of the moderated linear regressions in which COVID-19 precautions were separately regressed on the individual suppressor variables. We have also added prose to the Results section that contextualizes this figure, and explains how this analysis connects the suppressor variables to the relationship between socially conservative attitudes and COVID-19 precautions. See added prose below (lines 621-654)

In order to better visualize how these variables resulted in negative indirect effects between socially conservative attitudes and COVID-19 precautions, we regressed COVID-19 precautions on each suppressor variable, and their interactions with political party affiliation. The conditional effects were then plotted (Fig 2). In both studies, political party was a significant moderator of all four suppressor variables (see S3 Appendix for statistical details). In addition, greater trust in scientists, trust in liberals and moderates, and liberal media consumption were all positively correlated with COVID-19 precautions among Republicans and Independents. Greater economic conservatism was negatively correlated with COVID-19 precautions among Republicans and Independents (see S3 Appendix for statistical details). Further, in both Studies 1 and 2, socially conservative attitudes negatively associated with trust in scientists, trust in liberals and moderates, and economic liberalism among Republicans and Independents. Socially conservative attitudes negatively correlated with liberal media consumption among Republicans in both studies, but only among Independents in Study 2. See S3 Appendix for full details of these results. 

In sum, these results illustrate the pathways by which these four variables act as suppressors of a socially conservative attitudes-precautions relationship among Republicans and Independents. First, more socially conservative attitudes were negatively correlated with greater trust in scientists and liberal and moderate sources, and greater liberal media consumption, while being positively correlated with greater economic conservatism. Second, engaging in fewer COVID-19 precautions was associated with lower trust in scientists and liberal and moderate sources of information, and lesser liberal media consumption, while being positively associated with greater economic conservatism. Taken together, these relationships result in suppression of a positive relationship between greater socially conservative attitudes and greater COVID-19 precautions among Republicans and Independents.

.

R8 The authors wrote: 

“Thus, prior research on the traditional-norms account has not sufficiently taken into account the cost-benefit trade-offs of many threat and pathogen avoidance behaviors, in turn limiting the ecological validity of the findings.” 

This is a turning point in the introduction that leads to one of the central premises of their hypotheses, and it highlights the magnitude of the present contribution. As it stands right now, the text seems a bit vague and overly general. To enhance the strength of their argument, I would recommend the authors directly cite the work they are alluding to here and incorporate any other studies that have already tried to address those limitations (successfully or not). I feel the paper would benefit from being as direct and concise as possible with regards to the work they are citing, constructively criticizing, and building upon. 

REPLY: We agree that it enhances our argument to be more specific here. We have therefore added citations to instances of the type of work to which we are alluding, as well as citations to papers that have started to address the limitations that we describe; see below (lines 165-172).

Thus, prior research (Hibbing et al., 2014; Jost et al., 2007; Terizzi et al., 2013; Tybur et al., 2016) has focused mostly on the benefits of threat avoidance, in turn limiting the ecological validity of the findings. Greater attention to costs is needed to more fully understand cost-benefit tradeoffs. Accordingly, recent work has started to address the effects of tradeoffs on threat and pathogen avoidance behaviors (Gul et al., 2021; Tybur et al., 2020). In addition, the previously discussed empirical observations of conservative shifts in response to real-world threats likely implicitly summarize the cost-benefit trade-off calculations that individuals may be making.

R9 The authors wrote: 

“Some of the precautions recommended or required by public health authorities interfere with engaging in traditional practices (e.g., social distancing precludes family gatherings, public sporting events, in-person religious services, etc.). Accordingly, two possibilities exist regarding the relationship between threat sensitivity, socially conservative attitudes, and COVID-19 precautions. On the one hand, if more threat-sensitive individuals focus on the precautions themselves, construing these as infringing on traditions, then they will report lower precautionary behavior than less threat-sensitive individuals–and, indeed, may view such behaviors as threats in themselves, endangering individual liberties. On the other hand, if more threat-sensitive individuals focus on the danger posed by COVID-19 more so than the conflict with various traditions entailed by precautionary behaviors, then they will report both greater precautionary behavior and greater valuation of traditions than will less threat-sensitive individuals.”

I really like this conceptualization, and it helps facilitate a more holistic understanding of the dynamics of social conservatism and threat-avoidance in a more complex way than one normally encounters (which is a simple linear, “cause and effect” way). One recommendation, however, would be to elaborate on the explanation of the first possibility, because it is more novel in political psychology. I recommend describing the second possibility first (because it is the simpler one), and then discussing the one about “focusing on the precautions themselves.” I recommend turning the 4-line sentence, into perhaps two shorter sentences for the purpose of clarity.

REPLY: We agree with this suggestion. Resultantly, we have re-ordered the two possibilities per the Reviewer’s recommendation, and expanded on the possibility concerning the content of the precautions themselves. See added prose below (lines 188-200).

Some of the precautions recommended or required by public health authorities interfere with engaging in traditional practices (e.g., social distancing precludes family gatherings, public sporting events, in-person religious services, etc.). Accordingly, two possibilities exist regarding the relationship between threat sensitivity, socially conservative attitudes, and COVID-19 precautions. On the one hand, if more threat-sensitive individuals focus on the danger posed by COVID-19 over and above the conflict with various traditions entailed by precautionary behaviors, then they will report both greater precautionary behavior and greater valuation of traditions than will less threat-sensitive individuals. On the other hand, highly threat-sensitive individuals may view such behaviors as threats in themselves, endangering individual liberties or economic prosperity. If more threat-sensitive individuals focus on the precautions themselves, construing these as infringing on traditions, then they will report lower precautionary behavior than less threat-sensitive individuals, potentially resulting in a negative relationship between traditionalism and precautionary behaviors. 

R10 typo: “In” after last sentence of section. 

REPLY: Thank you, this typo has been corrected. 

R11 All the measures included here should be presented in the precise order that they were presented to participants in the study. Also, is the order of measures consistent across studies? 

REPLY: We have made sure that measures are presented in the manuscript in the same order that they were presented to participants, as well as specifying where the order of presentation was randomized (see lines 373-374). Finally, we have clarified in the manuscript that the order of measures was consistent across both studies (see line 373).

R12 I would recommend elaborating further on the development of the social conservative attitude scale, because it is the central variable of interest and the operationalization deviates somewhat from the more “traditional” or “common” conception and usage of ideology (the left-right ideological self-placement item). I suggest that the authors devote more space explaining and providing concrete examples about the development and operationalization of this variable in the text of the manuscript. 

REPLY: We have further elaborated on the development of the socially conservative attitudes scale, with particular attention paid to why we used this measure of ideology in comparison to a more typical self-placement item along a single left-right dimension, and further details on why we developed this particular measure. See added prose below (lines 401-419).

Political orientation is often measured using a single-item unidimensional scale ranging from conservative to liberal. However, as we have noted, it is critical to separate distinct dimensions of ideology (Claessens et al., 2020), such as economic and social conservatism or liberalism. Further, political ideology is complex, and encompasses both specific policy preferences in a given political context, as well as the kinds of general attitudes that help shape those preferences; in the context of social conservatism, the endorsement of tradition is likely a constituting attitude of the ideology. To operationalize social conservatism in light of these considerations—characterized by both specific policy preferences involving matters of tradition and cultural change, and general attitudinal orientation toward tradition and change—we created a composite socially conservative attitudes ideology scale. This composite scale consisted of the rescaled responses from the Dodd-style issues index and the conventionalism subscale of the ASC (see previous sections for example items). Both the issues index and the ASC scale have been widely used to measure social conservatism and attitudes toward tradition (e.g., Tybur et al., 2016; Dodd et al., 2012). Further, because these individual scales focus on, respectively, general attitudes toward tradition, and specific policy preferences related to social conservatism, combining them provides a more complete measurement of socially conservative ideology. The resultant composite socially conservative attitudes variable was measured on a -1-to-1 scale, where increasing scores indicate increasing socially conservative attitudes. This composite was reliable (αs = .89 – .90). 

R13 It would be useful if there were sample items provided, rather than a description of the contents of these items.

REPLY: We have added specific sample items to this section of the Methods, see lines 459-514.

R14 I would recommend using sub-headers for each of the 6 categories, to make the last section easier to follow. As it reads now, this passage may lead to a bit of confusion. For example, the authors write: 

(p. 17, line 382) The authors wrote:

“Third, participants were asked about the extent to which they were preparing for an economic downturn, such as by delaying major financial decisions. We averaged these behavior items into an economic precautions composite, which was reliable (αs = .75 – .78).”

I will recommend doing something like this: 

3. Economic precautions: Participants were asked about the extent to which they were preparing for an economic downturn by delaying major financial decisions. We averaged these behavioral items into a composite scale, which was reliable (αs = .75 – .78). 

REPLY: We agree that using subheaders makes this section easier to follow, and have instituted this change; see lines 459-514.

R15 It would be useful to remind the reader of the total sample size and report the precise number of self-identified “Other” parties.

REPLY: We have provided a reminder of the total sample size, and the precise number of participants who identified in other parties, see lines 538-541.

R16 (p19, line 423 and throughout the results section) Many of the reported statistics (e.g., p = 3.33e-6) do not conform to APA style.

REPLY: Thank you for catching this error, we have reformatted the Results section to conform to APA and PLOS One style requirements.

R17 They report: 

“Finally, in both studies, the interactions between political party affiliation and socially conservative attitudes were significant between Democrats on the one hand, and both Republicans (Study 1: B = -0.80, SE = .29, t(857) = -2.77, p = .006; Study 2: B = -0.79, SE = .26, t(854) = -3.04, p = .002) and Independents (Study 1: B = -0.69, SE = .25, t(857) = -2.72, p = .007; Study 2: B = -0.78, SE = .23, t(854) = -3.39, p = .001) on the other. Slopes did not significantly differ between Independents and Republicans (Study 1: B = 0.11, SE = .31, t(857) = .35, p = .728; Study 2: B = 0.01, SE = .28, t(854) = .04, p = .969).” 

It is not clear what are they trying to say with this. It seems like the only thing they should report here is that the slopes between Independents and Republicans did not differ from each other. The same is true on p. 22 (line 499). I recommend clarifying these points. 

REPLY: Our intention in this section was to report whether the interactions from the moderated linear regressions in which we regressed COVID-19 precautions on socially conservative attitudes and political party affiliation were significant. However, given that the differences between Democrats on the one hand, and Republicans and Independents on the other, are clearly depicted in Figure 1, we agree that it streamlines the section to only report the comparisons between the slopes of Independents and Republicans (see lines 567-569). 

In regard to the similar analyses reported starting at line 499 in the original manuscript, our intention was to illustrate that accounting for the suppressor variables resulted in slopes that did not significantly differ between Democrats on the one hand, and Republicans and Independents on the other, in contrast to the significant interactions observed without the suppressors. We have reformatted the presentation of these results in a way that hopefully makes our intention clearer; see lines 684-689. 

R18 The authors reported combined statistics like this: 

“We then performed simple slopes analyses, finding that, sensibly, in both studies, disgust sensitivity associated with precautionary behaviors among Democrats (Bs = .20 – .24, ps = 7.60e-7 – 2.66e-5), Republicans (Bs = .23  – .36, ps = 2.40e-8 – 4.76e-4), and Independents (Bs = .27 – .40, ps = 5.63e-11 – 1.16e-5).” 

I would recommend reporting these statistics separately for each study in a consistent way throughout the Results section. 

REPLY: We agree that the statistics should be presented in a uniform manner in the Results section, so we have reformatted those sections that reported combined statistics to be consistent with the rest of the section. See the revised prose below, lines 684-689, 733-738. 

Further, after accounting for the suppressors, in both studies, slopes did not significantly differ between Democrats and Republicans (Study 1: B = -.03, SE = .30, t(820) = -.10, p = .918; Study 2: B = -.33, SE = .25, t(812) = -1.34, p = .182), Democrats and Independents (Study 1: B = -.06, SE = .27, t(820) = -.24, p = .812; Study 2: B = .003, SE = .24, t(812) = -.01, p = .991), or Republicans and Independents (Study 1: B = -.03, SE = .32, t(820) = -.10, p = .920; Study 2: B = .33, SE = .27, t(812) = 1.21, p = .229).

We then performed simple slopes analyses, finding that, sensibly, in both studies, disgust sensitivity associated with precautionary behaviors among Democrats (Study 1: B = .20, SE = .05, t(856) = 4.22, p < .001; Study 2: B = .24, SE = .05, t(854) = 4.98, p < .001), Republicans (Study 1: B = .36, SE = .06, t(856) = 5.63, p < .001; Study 2: B = .23, SE = .07, t(854) = 3.51, p < .001), and Independents (Study 1: B = .40, SE = .06, t(856) = 6.64, p < .001; Study 2: B = .27, SE = .06, t(854) = 4.41, p < .001).

R19: They wrote: 

“[…] Nevertheless, in both studies, we found significant evidence that suppression by the target variables occurred, such that, when accounting for the suppressors, political party no longer moderated the relationship between socially conservative attitudes and precautions.”

The last sentence in the first limitations paragraph seems a bit contradictory to the main points they are raising just before. I recommend to simply take it out. 

REPLY: We agree with this recommendation; this sentence has been removed.

R20 They wrote: 

“Second, because of the cross-sectional, correlational, non-experimental design of the research, it is impossible to draw definitive conclusions regarding causal relationships between the phenomena of interest.” 

This statement seems pretty vague, and it would be beneficial to suggest ways of overcoming this limitation. Perhaps elaborate a bit more on the specific causal relationship you would like to address and report ways that one could potentially do so with different types of data or research designs. 

REPLY: We agree that this limitation is vague in the original manuscript. Therefore, we have added the following prose to the manuscript in order to elaborate on which causal relationships we would like to address, and how we might do so in future studies (lines 856-874):

Second, because of the cross-sectional, correlational, non-experimental design of the research, it is impossible to draw definitive conclusions regarding causal relationships between the phenomena of interest. However, future research could in part address these limitations. For example, in regard to the relationship between socially conservative attitudes and the suppressor variables, longitudinal research might investigate the extent to which partisan media environments shape beliefs and behaviors regarding different threats, versus the extent to which individuals seek out media environments that accord with their previous beliefs. Further, especially given the question of social conservative shifts during periods of threat, the direction of causality for the relationship between socially conservative attitudes and COVID-19 precautions found in this study is undetermined. For example, our results are consistent with the possibility of a socially conservative shift among Democrats most threatened by the pandemic; we cannot disentangle that causal pathway from one in which Democrats who were more socially conservative to begin with responded to the pandemic with greater threat avoidance. Again, longitudinal research may provide leverage on this issue. Alternatively, experimental elicitations of threat could also probe the question of directionality. However, it would be difficult to build the real-world contextualizing factors found in this study into an experimental design, thus limiting the inferential value. Finally, society-level data on the relationship between traditionalism and COVID-19 precautions may also shed light on the underlying causal relationships between the variables of interest.

R21 In sum, the inconsistencies with regards to the organization and content of the results section made the paper somewhat difficult to follow and comprehend. It was difficult to get a clear picture from the Results section in the main text alone, and without the supplemental material it was difficult to get a full picture of the results, as well as the analytical strategies implemented. I strongly recommend that these issues are resolved prior to publication. 

REPLY: Again, we’d like to thank the Reviewer for their excellent comments and recommendations. We think that the above changes resolve these issues, particularly in regard to the clarity of the Results section.

---

## [Editor Report · Decision Letter 1]

3 Jun 2021

Of Pathogens and Party Lines: Social Conservatism Positively Associates with COVID-19 Precautions among U.S. Democrats but not Republicans

PONE-D-21-04873R1

Dear Dr. Samore,

We’re pleased to inform you that your manuscript has been judged scientifically suitable for publication and will be formally accepted for publication once it meets all outstanding technical requirements.

Kind regards,

Valerio Capraro

Academic Editor

PLOS ONE
---

## [Editor Report · Acceptance letter]

18 Jun 2021

PONE-D-21-04873R1 

Of pathogens and party lines: Social conservatism positively associates with COVID-19 precautions among U.S. Democrats but not Republicans 

Dear Dr. Samore:

I'm pleased to inform you that your manuscript has been deemed suitable for publication in PLOS ONE. Congratulations! Your manuscript is now with our production department. 

Kind regards, 

on behalf of

Dr. Valerio Capraro 

Academic Editor

PLOS ONE